# MULTI-OBJECTIVE MODEL SELECTION FOR TIME SERIES FORECASTING

## ABSTRACT

Research on time series forecasting has predominantly focused on developing methods that improve accuracy. However, other criteria such as training time or latency are critical in many real-world applications. We therefore address the question of how to choose an appropriate forecasting model for a given dataset among the plethora of available forecasting methods when accuracy is only one of many criteria. For this, our contributions are two-fold. First, we present a comprehensive benchmark, evaluating 7 classical and 6 deep learning forecasting methods on 44 heterogeneous, publicly available datasets. The benchmark code is open-sourced along with evaluations and forecasts for all methods. These evaluations enable us to answer open questions such as the amount of data required for deep learning models to outperform classical ones. Second, we leverage the benchmark evaluations to learn good defaults that consider multiple objectives such as accuracy and latency. By learning a mapping from forecasting models to performance metrics, we show that our method PARETOSELECT is able to accurately select models from the Pareto front — alleviating the need to train or evaluate many forecasting models for model selection. To the best of our knowledge, PARETOSELECT constitutes the first method to learn default models in a multi-objective setting.

## 1 INTRODUCTION

For decades, businesses have been using time series forecasting to drive strategic decision-making (Simchi-Levi et al., 2013; Hyndman & Athanasopoulos, 2018; Petropoulos et al., 2020). Analysts leverage forecasts to gauge resource requirements, retailers forecast future product demand to optimize their supply chains, cloud providers predict future web traffic to scale server fleets, and in the energy sector, forecasting plays a crucial role e.g. to predict load and energy prices. In domains like these and many other, more precise predictions directly translate into an increase in profit. Thus, it is no surprise that research on forecasting methods has historically focused on improving accuracy.

In addition to more classical *local* forecasting methods which fit a model per time series, *global* forecasting models such as deep learning and tree-based models have demonstrated state-of-the-art forecasting accuracy (Wen et al., 2017; Oreshkin et al., 2019; Salinas et al., 2020a; Smyl, 2020) when sufficient training data is available (Makridakis et al., 2018; Januschowski et al., 2020). This research has led to a large variety of different forecasting models and hyperparameter choices (Hutter et al., 2019), where different models exhibit vastly different characteristics, including accuracy, training time, model size, and inference latency.

While this variety of forecasting models is a great resource, it introduces challenging questions. On the one hand, researchers would like to understand patterns in the performance of different models and to benchmark new models against existing methods. On the other hand, practitioners are interested in understanding which model performs best for a particular dataset or application.

In this work, we address both of these problems: we release one of the most comprehensive publicly available evaluations of forecasting models across 44 datasets. Using this benchmark dataset, we develop a novel method for learning *good defaults* for forecasting models on previously unseen datasets. Importantly, we adopt a multi-objective perspective that allows us to select forecasting models that are simultaneously accurate and satisfy constraints such as inference latency, training time, or model size.

The main contributions of this work can be summarized as follows:

- We release the evaluations of 13 forecasting methods on 44 public datasets with respect to multiple performance criteria (different forecast accuracy metrics, inference latency, training time, and model size). This constitutes, by far, the most comprehensive publicly available evaluation of forecasting methods. Those evaluations can be leveraged, for example, to assess the relative performance of future forecasting methods with little effort.

- As an example application of this benchmark, we use the data to perform a statistical analysis which shows that only a few thousands observations are required for deep learning methods to outperform classical methods. In addition, we investigate the benefit of ensembling forecasting models.

- We propose a novel method that can leverage offline evaluations to learn good default models for unseen datasets. Default models are selected to optimize for multiple objectives.

- We introduce a technique for ensembling models in a multi-objective setting where the resulting ensemble is not only highly accurate but also optimized for other objectives, such as a low inference latency.

## 2 RELATED WORK

**Time Series Forecasting.**   Time series forecasting has seen a recent surge in attention by the academic community that has started to reflect its relevance in business applications. Traditionally, univariate, so-called *local* models that consider time series individually have dominated (Hyndman & Athanasopoulos, 2018). However, in modern applications, methods that learn *globally* across a set of time series can be more accurate (Oreshkin et al., 2019; Salinas et al., 2020a; Lim et al., 2021; Montero-Manso & Hyndman, 2021) — in particular, methods that rely on deep learning. With a considerable choice of models available, it is unclear which methods should perform best on which forecasting dataset, unlike in other machine learning domains where dominant approaches exist.

In domains such as computer vision or neural architecture search (NAS), offline computations have been harvested and leveraged successfully to perform extensive model comparisons or compare different NAS strategies (Ying et al., 2019; Dong & Yang, 2020; Pfisterer et al., 2021). However, to the best of our knowledge, no comprehensive set of model evaluations has been released in the realm of forecasting. Consequently, some questions remain open: how much data is required for global deep learning methods to outperform classical local methods such as ARIMA or ETS (Hyndman & Athanasopoulos, 2018)? What is the impact on accuracy when ensembling different forecasting models? While some recent methods incorporate ensembling (Oreshkin et al., 2019; Jeon & Seong, 2021), comparisons are often made against individual models which may cloud the benefit of the method proposed versus the sole benefit of ensembling.

**Learning Default Models.**   Finding the best model or set of hyperparameters is often performed via Bayesian optimization given its theoretical regret guarantees (Srinivas et al., 2012). However, even with early-stopping techniques (Golovin et al., 2017; Li et al., 2017), practitioners often restrict the search to a single model due to the large cost of training many models. One technique to drastically speed up the model/hyperparameter search is to reuse offline evaluations of related datasets via transfer learning. For instance, Wistuba et al. (2015); Winkelmolen et al. (2020); Pfisterer et al. (2021) leverage offline evaluations to alleviate the need for training many models by learning a small list of $n$ good defaults that provide a small joint error when evaluated on all datasets. This approach bears a lot of similarity with ours. Key differences are that we consider multiple objectives and model ensembles. In the time series domain, Shah et al. (2021) also considers the task of automatically choosing from a large pool of forecasting models the one performing best on a particular dataset but also only optimizes for a single objective and relies on potentially expensive model training for model selection.

## 3 BENCHMARKING FORECASTING METHODS

In this section, we formally introduce the problem of time series forecasting and subsequently provide an overview of the benchmark evaluations that we release. By evaluating 13 forecasting meth-

ods along with different hyperparameter choices on all 44 benchmark datasets and for two random seeds, the benchmark comprises 4,708 training runs and amasses over 1 TiB of forecast data. At the end of this section, we demonstrate how this benchmark data can be used to perform an extensive comparison of contemporary forecasting methods.

## 3.1 TIME SERIES FORECASTING

The goal of time series forecasting is to predict the future values of one or more time series based on historical observations. Formally, we consider a set $\mathcal{Z} = \{\boldsymbol{z}_{1:T_i}^{(i)}\}_{i=1}^{K}$ of $K$ univariate, equally-spaced time series of lengths $T_i$, where $\boldsymbol{z}_{a:b}^{(i)} = (z_a^{(i)}, z_{a+1}^{(i)}, \ldots, z_b^{(i)})$ denotes the vector of observations for the $i$-th time series in the time interval $a \leq t \leq b$. In probabilistic time series forecasting, a model then estimates the probability distribution across the forecast horizon $\tau$, i.e. the distribution over the $\tau$ future values, from the historical observations:

$$\mathbb{P}\left(\boldsymbol{z}_{T_{i+1}:T_{i+\tau}}^{(i)} \,\middle|\, \boldsymbol{z}_{1:T_i}^{(i)}\right) \tag{1}$$

Models typically approximate the joint distribution of Eq. (1) via Monte Carlo sampling (Salinas et al., 2020a) or learn to directly predict a set of distribution quantiles for multiple time steps using quantile regression (Wen et al., 2017; Lim et al., 2021).

## 3.2 METHODS

We consider 13 models in total that can be distinguished into 7 *local* methods (estimating parameters individually from each time series) and 6 *global* methods (learning from all available time series jointly). Details about models can be found in Appendix A.

**Local Methods.** We use *Seasonal Naïve* (Hyndman & Athanasopoulos, 2018) and *NPTS* (Rangapuram et al., 2021) as simple, non-parametric baselines. Further, we consider *ARIMA* and *ETS* (Hyndman & Athanasopoulos, 2018) as well-known statistical methods and *STL-AR* (Talagala, 2021) and *Theta* (Assimakopoulos & Nikolopoulos, 2000) as inexpensive alternatives. Lastly, we include *Prophet* (Taylor & Letham, 2018), an interpretable model that has received plenty of attention.

**Global Methods.** All global methods that we use are deep learning models, namely: *Simple Feedforward* (Alexandrov et al., 2020), *MQ-CNN* and *MQ-RNN* (Wen et al., 2017), *DeepAR* (Salinas et al., 2020a), *N-BEATS* (Oreshkin et al., 2019), and *TFT* (Lim et al., 2021). For each model (except MQ-RNN), we consider three hyperparameter settings: this includes the default set provided by their implementation as well as hyperparameter sets that roughly halve and double the default model capacity (see Table 2 in the appendix). Additionally, we consider three different context lengths that govern the length of the time series that predictions are conditioned on.

**Model Training.** Model training is only required for deep learning models since parametric local methods estimate parameters at prediction time. Deep learning models are trained for a fixed duration depending on the size of the dataset. Further details can be found in Appendix C.

## 3.3 DATASETS

Our benchmark provides 44 heterogeneous public datasets in total. The datasets greatly differ in the number of time series (from 8 to $\approx$170,000), their mean length (from $\approx$22 to $\approx$500,000), their frequency (minutely, hourly, daily, weekly, monthly, quarterly, yearly), and the forecast horizon (from 6 to 60). Thus, we expect them to cover a wide range of datasets encountered in practice.

Datasets are taken from various forecasting competitions, the UCI (Dua & Graff, 2017), and the Monash time series forecasting repository (Godahewa et al., 2021). Dataset sources, descriptions, basic statistics, and an explanation of the data preparation procedure can be found in Appendix B.

## 3.4 METRICS

To measure the accuracy of forecasting models, we employ the normalized *continuous ranked probability score* (nCRPS) (Matheson & Winkler, 1976; Gneiting & Raftery, 2007) whose definition

Table 1: The average performance of various forecasting methods across all datasets with respect to relative latency (compared to Seasonal Naïve) and nCRPS. The table shows classical methods (top), deep learning methods (middle, left), hyper-ensembles using all hyperparameter configurations of deep learning models (middle, right), and latency-constrained ensembles (bottom). Best values across within each group are displayed in bold, best values across all models are starred.

| | Rel. Latency | nCRPS Rank |
|---|---|---|
| **ARIMA** | $31100.33 \pm {}_{11739.26}$ | $\mathbf{13.86} \pm \mathbf{5.83}$ |
| **ETS** | $1183.26 \pm {}_{321.88}$ | $15.45 \pm {}_{6.87}$ |
| **NPTS** | $133.64 \pm {}_{20.82}$ | $17.82 \pm {}_{7.35}$ |
| **Prophet** | $2635.38 \pm {}_{393.48}$ | $17.82 \pm {}_{5.52}$ |
| **Seasonal Naïve** | $\mathbf{*1.00} \pm \mathbf{0.00}$ | $19.86 \pm {}_{4.22}$ |
| **STL-AR** | $76.12 \pm {}_{7.44}$ | $15.57 \pm {}_{6.73}$ |
| **Theta** | $25.15 \pm {}_{1.55}$ | $15.59 \pm {}_{6.19}$ |
| **DeepAR** | $21.55 \pm {}_{2.33} / 310.62 \pm {}_{34.26}$ | $9.91 \pm {}_{5.79} / \mathbf{6.73} \pm \mathbf{5.24}$ |
| **MQ-CNN** | $1.78 \pm {}_{0.15} / 17.48 \pm {}_{1.60}$ | $15.86 \pm {}_{5.75} / 13.07 \pm {}_{5.98}$ |
| **MQ-RNN** | $2.01 \pm {}_{0.18} / \mathbf{6.53} \pm \mathbf{0.63}$ | $22.07 \pm {}_{4.30} / 21.02 \pm {}_{4.92}$ |
| **N-BEATS** | $3.33 \pm {}_{0.21} / 34.29 \pm {}_{2.25}$ | $18.32 \pm {}_{3.45} / 16.59 \pm {}_{3.71}$ |
| **Simple Feedforward** | $\mathbf{1.57} \pm \mathbf{0.07} / 14.12 \pm {}_{0.63}$ | $12.32 \pm {}_{4.24} / 9.45 \pm {}_{4.16}$ |
| **TFT** | $4.66 \pm {}_{0.31} / 44.08 \pm {}_{3.06}$ | $\mathbf{8.70} \pm \mathbf{4.71} / 6.75 \pm {}_{4.61}$ |
| **Constrained Ensemble (1 ms)** | $\mathbf{2.29} \pm \mathbf{0.16}$ | $13.77 \pm {}_{5.01}$ |
| **Constrained Ensemble (5 ms)** | $11.99 \pm {}_{1.09}$ | $9.45 \pm {}_{5.20}$ |
| **Constrained Ensemble (10 ms)** | $24.20 \pm {}_{2.18}$ | $8.30 \pm {}_{4.97}$ |
| **Constrained Ensemble (50 ms)** | $107.68 \pm {}_{9.44}$ | $6.57 \pm {}_{5.04}$ |
| **Constrained Ensemble (100 ms)** | $157.75 \pm {}_{14.86}$ | $5.55 \pm {}_{4.28}$ |
| **Unconstrained Ensemble** | $196.41 \pm {}_{20.44}$ | $\mathbf{*4.36} \pm \mathbf{3.37}$ |

is given in Appendix A. The benchmark dataset that we release contains four additional forecast accuracy metrics (MAPE, sMAPE, NRMSE, ND). Besides forecasting accuracy metrics, we store inference latency, model size (i.e. number of parameters) and training time for deep learning models. Latency is measured by dividing the time taken to generate predictions for all test time series in the dataset by their number.

## 3.5 BENCHMARK ANALYSIS

Having outlined the benchmark, we now want to demonstrate the usefulness of the evaluations for research. For this, we want to analyze how local (classical) and global (deep learning) forecasting methods compare against each other. In fact, this comparison has been the object of heated discussions in the forecasting community (Makridakis et al., 2018; Januschowski et al., 2020).

**Method Comparison.** In order to compare methods, we consider their latency and accuracy (in terms of nCRPS) across all datasets. Table 1 shows each method's relative latency compared to Seasonal Naïve as well as the the methods' nCRPS rank[1]. As far as latency is concerned, global methods, once trained, allow to generate forecasts considerably faster than local methods (except for Seasonal Naïve). The reason is simple: once trained, they simply need to run a forward pass. In contrast, the implementations for the local methods chosen here do not differentiate between training and inference, but rather estimate their parameters at prediction time. Across datasets, the relative latency of all considered deep learning models improves between 15% and 95% upon the latency of the fastest local method (excluding Seasonal Naïve). As far as accuracy is concerned, complex deep learning models — namely DeepAR and TFT — generally perform best across the benchmark datasets. Except for MQ-RNN, global methods compare favorably against local methods.

As a measure of *relative* model stability, Table 1 additionally shows the standard deviation of the nCRPS rank across all benchmark datasets. Not only is TFT the method which performs best on average, its rank is also comparatively consistent compared to other methods. Especially,

---

[1]Ranks are computed over all methods and ensembles, including the constrained ensembles of Section 4.4.

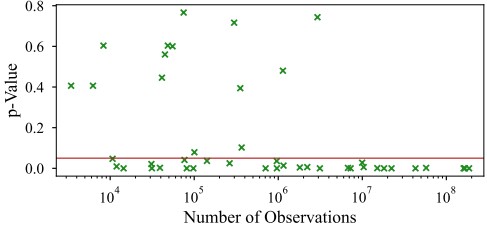 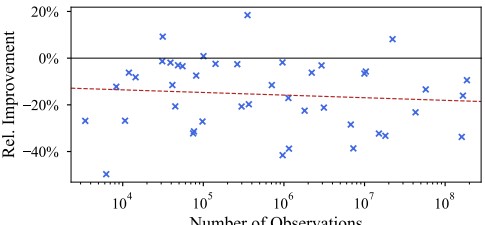

Figure 1: The $p$-values of the null hypothesis that statistical methods perform equal to or better than deep learning methods, evaluated for datasets of different sizes. The red line displays the significance level $\alpha = 5\%$.

Figure 2: The relative improvement of the best deep learning method versus the best classical method on all benchmark datasets. The red line is fitted on the visualized datapoints via linear regression in the log-space.

Table 1 also indicates the performance of ensembles obtained by averaging the predictions of $n = 9$ ($n = 3$ for MQ-RNN) deep learning models trained with different hyperparameters (*hyper-ensembles*) as done for N-BEATS (Oreshkin et al., 2019) or in M5 competition's third-ranking method (Jeon & Seong, 2021). While generally bearing a significant cost in latency, the benefit of ensembling is significant. For instance, the Simple Feedforward hyper-ensemble yields a competitive model that outperforms DeepAR in terms of both accuracy and latency. Lastly, DeepAR and TFT hyper-ensembles result in the most competitive ensembles.

**Comparison of Classical and Deep Learning Methods.** We further conduct a statistical analysis to compare the classical time series models with the deep learning models listed in Section 3.2. For this, we construct the null hypothesis that the accuracy of classical models is equal to or better than the accuracy of deep learning methods. This is a claim put forward, for example, by Makridakis et al. (2018). More formally, we write $H_0 : \mathcal{Q}_{\text{class}} \leq \mathcal{Q}_{\text{deep}}$ where $\mathcal{Q}_{\text{class}}$ and $\mathcal{Q}_{\text{deep}}$ describe the distribution of nCRPS values of classical and deep learning models, respectively. Samples of the distributions are derived from the respective single-model evaluations in our benchmark. $\mathcal{Q}_{\text{class}}$ and $\mathcal{Q}_{\text{deep}}$ are continuous and belong to unknown families of distributions with unequal variances. Hence, we choose the nonparametric two-sample Kolmogorov-Smirnov test (Gibbons & Chakraborti, 2003) to compute the $p$-value, i.e. the probability of $H_0$ holding true, on each individual dataset.

We evaluate the null hypothesis $H_0$ on all 44 benchmark datasets and plot the $p$-value for the different dataset sizes in Figure 1. At a significance level of $\alpha = 5\%$, $H_0$ can be rejected for 30 out of 44 datasets. In fact, deep learning models exhibit competitive performance for almost all datasets and regularly outperform classical methods. Further, they tend to perform comparatively better with increasing dataset size — even if for some large datasets, classical models are indistinguishable — and, nonetheless, show competitive performance for small datasets. The latter is of particular interest since it contradicts tribal knowledge that large datasets are needed to outperform local/classical methods and disputes the claim presented by Makridakis et al. (2018) since only a few thousands observations seem sufficient to outperform the classical models considered.

These statements are further supported by Figure 2 where we compare the best deep learning model $x_{\text{deep}}$ against the best classical model $x_{\text{class}}$ on each dataset. We compute the relative improvement given by deep learning models as $\frac{\text{nCRPS}(x_{\text{deep}})}{\text{nCRPS}(x_{\text{class}})} - 1$. Except for few outliers, the best deep learning model outperforms all classical models on 40 out of 44 datasets (see Appendix F for a further analysis). Fitting a linear regression model on the relative improvements shows that the improvement by using deep learning models is only slightly amplified as the datasets grow in size.

We note, that we conflate deep learning and global models in our above discussion. Montero-Manso & Hyndman (2021) show the general superiority of global models over local models both theoretically and empirically and our results confirm their findings further.

## 4    LEARNING MULTI-OBJECTIVE DEFAULTS

The benchmark introduced in the previous section outlined how to acquire offline evaluations of forecasting methods on various datasets. However, in the presence of multiple conflicting objectives

such as accuracy and latency, it is not clear how one could choose the best models. In this section, we first formalize the problem of learning good defaults from these evaluations for a single objective. Afterwards, we formally introduce multi-objective optimization and show how learning defaults can be extended to account for multiple objectives. In the end, we report experimental results obtained by using our method.

## 4.1 PREREQUISITES

In our setting, we consider an objective function $\boldsymbol{f} : \mathcal{X} \to \mathbb{R}^m$ for a given task (in our case, a task is a dataset). The objective function maps any time series model $x \in \mathcal{X}$ to $m$ objectives (such as nCRPS, latency, etc.) that ought to be minimized. In addition, we assume that model evaluations on $T$ different but related tasks with objective functions $\boldsymbol{f}^{(1)}, \dots, \boldsymbol{f}^{(T)}$ are available. The set of offline model evaluations is then given as

$$\mathcal{D} = \bigcup_{j=1}^{T} \left\{ \left( x_i^{(j)}, \boldsymbol{y}_i^{(j)} \right) \right\}_{i=1}^{N_j} \qquad \text{with} \quad \boldsymbol{y}_i^{(j)} = \boldsymbol{f}^{(j)}(x_i^{(j)}) \in \mathbb{R}^m \qquad (2)$$

where $N_j$ evaluations are available for task $j$ and $\boldsymbol{y}_i^{(j)}$ denotes the evaluation of $x_i^{(j)}$ on task $j$.

**Learning Defaults.** To learn a set of $n$ good default models for unseen datasets in the single-objective setting, Pfisterer et al. (2021) propose to pick the models that minimize the joint error obtained when evaluating them on all datasets. This amount to pick the set of models $\{x_1, \dots, x_n\} \subset \mathcal{X}$ which minimize the following objective:

$$\min_{\{x_1, \dots, x_n\} \subset \mathcal{X}} \sum_{j=1}^{T} \min_{1 \le i \le n} f_k^{(j)}(x_i) \qquad (3)$$

where $k$ denotes a single fixed objective (for instance, the classification error). Since the problem is NP-complete, a greedy heuristic with a provable approximation guarantee is used: it iteratively adds the model minimizing Eq. 3 to the current selection. However, this approach only works for a single objective of choice $f_k$ rather than taking all objectives $\boldsymbol{f}$ (such as accuracy and latency) into account. In addition, it does not easily support the selection of ensembles since objectives can interact in differently when models are ensembled: while accuracy will likely increase by a small amount, the latencies of ensemble members need to be added up. Taking into account all possible ensembles of a given size $n$ would also blow up the combinatoric search when optimizing Eq. 3.

**Multi-Objective Optimization.** In multi-objective optimization, there generally exists no singular solution. Thus, when considering the objective function $\boldsymbol{f}$, there exists no single optimal model, but rather a set of optimal models (Emmerich & Deutz, 2018). A model $x \in \mathcal{X}$ dominates another model $x' \in \mathcal{X}$ ($x \prec_{\boldsymbol{f}} x'$) if $f_i(x) \le f_i(x')$ for all $i$ and there exists some $i$ such that $f_i(x) < f_i(x')$. The set of all non-dominated solutions (i.e. models) is denoted as the Pareto front $\mathcal{P}_{\boldsymbol{f}}(\mathcal{X})$ whose members are commonly referred to as Pareto-optimal solutions:

$$\mathcal{P}_{\boldsymbol{f}}(\mathcal{X}) = \{ x \in \mathcal{X} \mid \neg \exists x' \in \mathcal{X} : x' \prec_{\boldsymbol{f}} x \} \qquad (4)$$

To quantify the quality of a set $\mathcal{S} = \{x_1, \dots, x_n\} \subset \mathcal{X}$ of selected models, we use the hypervolume error (Zitzler & Thiele, 1998; Li & Yao, 2019). Given a set of points $\mathcal{Y} \subset \mathbb{R}^m$ and a reference point $\boldsymbol{r} \in \mathbb{R}^m$, we first define the hypervolume as the Lebesgue measure $\Lambda$ of the dominated space:

$$\mathcal{H}(\mathcal{Y}) = \Lambda \left( \{ \boldsymbol{q} \in \mathbb{R}^m \mid \exists \boldsymbol{p} \in \mathcal{Y} : \boldsymbol{p} \le \boldsymbol{q} \wedge \boldsymbol{q} \le \boldsymbol{r} \} \right) \qquad (5)$$

In turn, the hypervolume error $\varepsilon$ of the set $\mathcal{S}$ is defined as the difference in hypervolume compared to the true Pareto front $\mathcal{P}_{\boldsymbol{f}}(\mathcal{X})$:

$$\varepsilon(\mathcal{S}) = \mathcal{H} \left( \{ \boldsymbol{f}(x) \mid x \in \mathcal{P}_{\boldsymbol{f}}(\mathcal{X}) \} \right) - \mathcal{H} \left( \{ \boldsymbol{f}(x_1), \dots, \boldsymbol{f}(x_n) \} \right) \ge 0 \qquad (6)$$

Thus, the hypervolume error $\varepsilon(\mathcal{S})$ reaches its minimum when $\mathcal{S} \supseteq \mathcal{P}_{\boldsymbol{f}}(\mathcal{X})$. To account for different scales in the optimization objectives, we further standardize all objectives by quantile normalization (Bolstad et al., 2003) such that they follow a uniform distribution $\mathcal{U}(0, 1)$ — this is feasible as $\mathcal{X}$ is finite. While the normalization makes our comparisons robust to monotonic transformations of the objectives (Binois et al., 2020), it also allows us to choose the reference point for Eq. 5 as $\boldsymbol{r} = \boldsymbol{1}_m$.

**Learning Multi-Objective Defaults.** Using the hypervolume error, we can extend the minimization problem of Eq. 3 to learn defaults in the multi-objective setting. For this, we propose the following minimization problem:

$$\min_{\{x_1,\ldots,x_n\} \subset \mathcal{X}} \sum_{j=1}^{T} \varepsilon(\{x_1,\ldots,x_n\}) \tag{7}$$

In words, we seek to find a set of complementary model configurations $\{x_1,\ldots,x_n\} \subset \mathcal{X}$ that provide a good approximation of the Pareto front $\mathcal{P}_{\boldsymbol{f}^{(j)}}(\mathcal{X})$ across tasks $j \in \{1,\ldots,T\}$.

## 4.2 PARETOSELECT

We now introduce PARETOSELECT which tackles the minimization of Eq. 7. On a high-level, PARETOSELECT works as follows: first, it fits a parametric surrogate model $\tilde{\boldsymbol{f}}_\theta$ that predicts the performances of model configurations by leveraging the evaluations of $\mathcal{D}$. Then, it uses the surrogate $\tilde{\boldsymbol{f}}_\theta$ to estimate the objectives for all models in $\mathcal{X}$ and applies the non-dominated sorting algorithm on the objectives to select a list of default models.

The surrogate model $\tilde{\boldsymbol{f}}_\theta$ is trained by attempting to correctly rank model configurations within each task in the available offline evaluations $\mathcal{D}$. For each model $x \in \mathcal{X}$, $\tilde{\boldsymbol{f}}_\theta$ outputs a vector $\tilde{y} \in \mathbb{R}^m$ for $m$ optimization objectives. Models can then be ranked for each objective independently. In order to fit $\tilde{\boldsymbol{f}}_\theta$ on the data $\mathcal{D}$, we use listwise ranking (Xia et al., 2008) and apply linear discounting to focus on correctly identifying the top configurations similar to Zügner et al. (2020). The corresponding loss function can be found in Appendix D.

For the parametric surrogate model $\tilde{\boldsymbol{f}}_\theta$, we choose a simple MLP similar to Salinas et al. (2020b). We use two hidden layers of size 32 with LeakyReLU activations after each hidden layer and apply a weight decay of 0.01 as regularization. Importantly, we do not tune these hyperparameters. Predictive performances for every model can eventually be computed as

$$\tilde{\mathcal{Y}} = \{\tilde{\boldsymbol{f}}_\theta(x) \mid x \in \mathcal{X}\} \tag{8}$$

We note that for minimizing Eq. 7, either regression or ranking objectives can be used: since we apply quantile normalization to compute the hypervolume, the magnitude of the values predicted by $\tilde{\boldsymbol{f}}_\theta$ is irrelevant. However, we find the ranking setting to be superior to regression for finding good default models (see Appendix D for more details).

**Multi-Objective Sorting.** The "best" configurations are then determined by applying the non-dominated sorting algorithm (NDSA) — an extension of sorting to the multi-dimensional setting (Srinivas & Deb, 1994; Emmerich & Deutz, 2018) — to the predictive performances $\tilde{\mathcal{Y}}$. Figure 3 provides an illustration of the sorting procedure. NDSA can be described as a two-level procedure. First, it partitions the configurations to be sorted in multiple layers by iteratively computing the Pareto fronts, then it applies a sorting function `sort` to rank the configurations in each layer:

$$X_1 = \texttt{sort}\left(\mathcal{P}_{\tilde{\boldsymbol{f}}_\theta}(\mathcal{X})\right), \quad X_{i+1} = \texttt{sort}\left(\mathcal{P}_{\tilde{\boldsymbol{f}}_\theta}\left(\mathcal{X} \setminus \bigcup_{j \leq i} X_j\right)\right) \tag{9}$$

We choose `sort` to compute an $\epsilon$-net (Clarkson, 2006) for which there exists a simple greedy algorithm. When sorting a set $X$, the $\epsilon$-net sorting operation yields an order $\texttt{sort}(X) = [x_1,\ldots,x_{|X|}]$. The first element $x_1$ is chosen randomly from $X$ and $x_{i+1}$ is defined iteratively as the item which is farthest from the current selection in terms of Euclidean distance, formally:

$$x_{i+1} = \text{argmax}_{x \in X} \min_{j \leq i} \|\boldsymbol{f}(x) - \boldsymbol{f}(x_j)\|_2 \tag{10}$$

The final ordering is eventually obtained by concatenating the sorted layers. While previous work followed different approaches for the `sort` operation in the NDSA algorithm (Srinivas & Deb, 1994; Deb et al., 2002), we leverage the $\epsilon$-net since Salinas et al. (2021) highlighted its theoretical guarantees and Schmucker et al. (2021) provided empirical evidence for its good performance.

The pseudo-code of PARETOSELECT is given in Algorithm 1 in the appendix. In the case where predictions of the surrogate are error-free, the recommendations of the algorithm can be guaranteed to be perfect with zero hypervolume error.

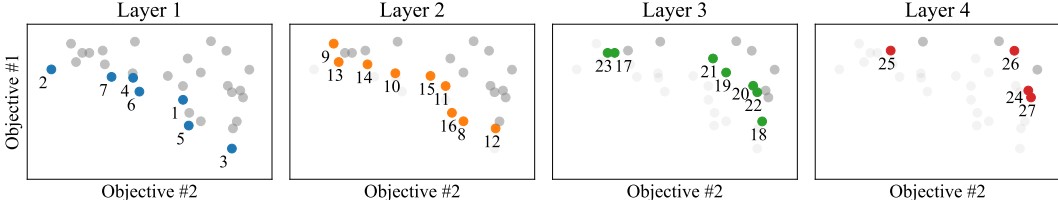

Figure 3: Illustration of non-dominated sorting. The layers show the partitioning of the data in Pareto fronts. The numbers depict the overall rank by computing the $\epsilon$-net within each layer.

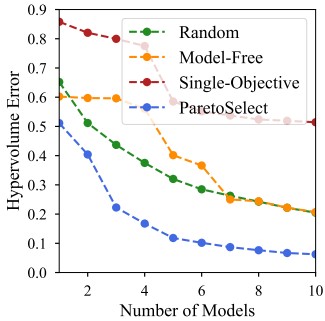

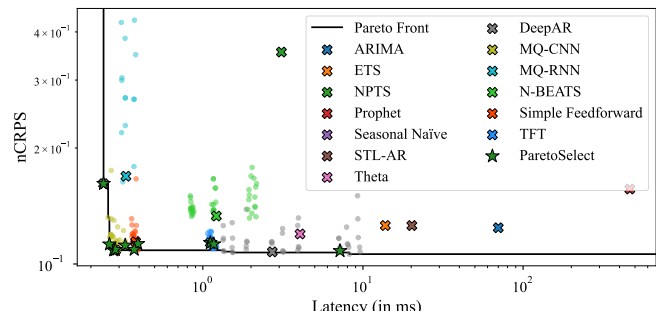

Figure 4: Hypervolume error when iteratively choosing models via different model selection approaches. Errors are averaged over all 44 benchmark datasets and five random seeds.

Figure 5: Visualization of forecast latency and nCRPS for models trained and evaluated on the "M4 Yearly" dataset. Each color describes a family of models with dots representing different hyperparameter configurations and training times (checkpoints for deep learning models). Crosses show the default hyperparameter configuration and the maximum training time for the dataset.

**Proposition 1.** *Assume that* $n \geq |\mathcal{P}_{\boldsymbol{f}}(\mathcal{X})|$ *and for all* $x \in \mathcal{X}$, $\tilde{\boldsymbol{f}}_{\theta}(x) = \boldsymbol{f}(x)$. *Then,* $\varepsilon(\{x_1, \ldots, x_n\}) = 0$ *where* $x_1, \ldots, x_n$ *are the first* $n$ *models selected by our method.*

*Proof.* By Eq. 9, the first $k$ recommendations of Pareto Select are such that $\{x_1, \ldots, x_k\} = \mathcal{P}_{\tilde{\boldsymbol{f}}_{\theta}}(\mathcal{X})$ where $k = |\mathcal{P}_{\tilde{\boldsymbol{f}}_{\theta}}(\mathcal{X})|$. Since for all $x \in \mathcal{X}$, $\tilde{\boldsymbol{f}}_{\theta}(x) = \boldsymbol{f}(x)$, we get that $\mathcal{P}_{\tilde{\boldsymbol{f}}_{\theta}}(\mathcal{X}) = \mathcal{P}_{\boldsymbol{f}}(\mathcal{X})$. It follows that for any $k \leq n$, $\mathcal{P}_{\boldsymbol{f}}(\mathcal{X}) = \{x_1, \ldots, x_k\} \subset \{x_1, \ldots, x_n\}$ and consequently $\varepsilon(\{x_1, \ldots, x_n\}) = 0$ since the hypervolume error of any set of points containing the Pareto front is zero. □

### 4.3 RESULTS

In order to evaluate our approach, we leverage the data collected with the benchmark presented in Section 3. For evaluation, we perform leave-one-out-cross-validation where we use each dataset one after the other as the test dataset and estimate the parameters $\theta$ of our parametric surrogate model $\tilde{\boldsymbol{f}}_{\theta}$ on the remaining 43 datasets. The multi-objective setting in the following experiments aims to minimize latency as well as nCRPS.

In Figure 4, we report the average hypervolume obtained for PARETOSELECT and several baselines, averaged over all 44 test datasets. PARETOSELECT picks the top elements of the non-dominated sort on the surrogate's predictive performances. As baselines, we consider a variant of PARETOSELECT which considers the nCRPS as the single objective (**Single-Objective**), the approach of Pfisterer et al. (2021) which greedily picks models to minimize the joint nCRPS on the training data (**Greedy Model-Free**), and random search over the entire model space of size $|\mathcal{X}| = 247$ (**Random**). The hypervolume obtained by PARETOSELECT with 10 default models is small ($\approx 0.06$) compared to the best baseline that reaches $\approx 0.20$. Further, random search requires a total of 27 models to be on par with the hypervolume of only 10 models selected via our method.

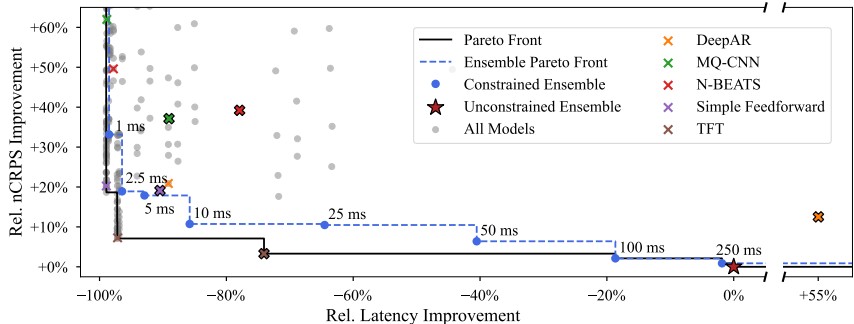

Figure 6: Comparison of individual forecasting models, hyper-ensembles and latency-constrained ensembles. The axes show the average relative latency and nCRPS compared to the unconstrained ensemble. Results are averaged across all 44 benchmark datasets. Light crosses show default configurations of deep learning models, bold crosses show their corresponding hyper-ensembles. Classical models are not colored as most can only be found (far) beyond the range of the plot.

Figure 5 depicts the top $n = 10$ recommendations produced by PARETOSELECT when predicting default models on a single dataset. The default models cover the true Pareto front well: our method is able to pick models with low latency, low nCRPS, and a good trade-off between these two objectives.

## 4.4 MULTI-OBJECTIVE ENSEMBLING

As shown in Section 3.5, ensembling models yields significant gains in accuracy but comes at the cost of considerable latency. Since a massively large ensemble model is of no practical use, it is critical to be able to select an ensemble with a latency that is acceptable for an application. Thus, we consider ensembles under different latency constraints.

For a latency constraint $c$, we build an ensemble with $n \leq 10$ members $x_1, \ldots, x_n$ by iteratively picking the model with best nCRPS such that the ensemble latency constrain $c$ is still met. To generate ensemble predictions, we combine the forecasts of the base models $x_1, \ldots, x_n$ by averaging all 10-quantiles $\{0.1, \ldots, 0.9\}$ for each time series and time step independently. Albeit the performances of the base models might differ significantly, our experiments showed that more complicated, non-uniform weighting schemes yielded less accurate forecasts on average. This aligns with current literature showing that simple averaging is often most effective (Petropoulos et al., 2020).

Figure 6 shows the performance of different latency-constrained ensembles, relative to the performance of an unconstrained ensemble that picks the $n = 10$ models with lowest predicted nCRPS. As expected, higher latency constraints $c$ allows the ensemble to perform better. For $c \geq 100$ ms, the constrained ensemble recovers the Pareto front and outperforms all individual models $x \in \mathcal{X}$ as well as ensembles. The small gap to the Pareto front for constraints $c \leq 50$ ms can be attributed to multiple effects. First, the nCRPS values predicted by the surrogate $\tilde{f}_\theta$ are not optimal. Second, models may not satisfy $c$ on every dataset although they have a lower latency on average (e.g. the default TFT configuration violates $c = 2.5$ ms on 17/44 datasets). Table 1 additionally reports performance metrics of latency-constrained ensembles: for instance, the ensemble for $c = 10$ ms outperforms all individual base models in terms of average nCRPS rank while the ensemble for $c = 100$ ms outperforms all hyper-ensembles as well.

## 5 CONCLUSION

In this work, we have presented (i) a new benchmark for evaluating forecasting methods in terms of multiple metrics and (ii) PARETOSELECT, a new algorithm that can learn default model configurations in a multi-objective setting. In future work, we will consider applying those techniques to other domains such as computer vision where offline evaluations of multiple datasets were recently made available by Dong & Yang (2020). Additionally, future research may extend the presented predictive surrogate model to the case of ensembles: this would enable PARETOSELECT to be used directly to select members for ensembles of any latency.

## 6 REPRODUCIBILITY

Public datasets and method implementations were used to ensure the reproducibility of our experiments. In Appendix A, we provide all hyperparameters used for training the benchmark forecasting models via GluonTS. The source and processing of each dataset as well as the procedure for generating data splits is detailed in Appendix B. We also provide details on the surrogate model used and a comparison with other choices in Appendix D. The pseudocode of PARETOSELECT is given in Appendix E. As a source of truth, we share the entire code used to run the benchmark and its evaluation with the submission. This code as well as the generated evaluations will be released with the paper.

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

Table 2: Overview of all models considered and their associated hyperparameters. The upper seven models do not require training. The lower six models are deep learning models. For all deep learning models, three context lengths are considered as well as three hyperparameter configurations (except for MQ-RNN) that result in small, medium and large models, respectively. Hyperparameters that are not augmented keep their default values. The hyperparameters of the "default" models correspond to the medium size.

| | Context Lengths | Hyperparameters | |
|---|---|---|---|
| **ARIMA** (Hyndman & Athanasopoulos, 2018) | — | — | |
| **ETS** (Hyndman & Athanasopoulos, 2018) | — | — | |
| **NPTS** (Rangapuram et al., 2021) | — | — | |
| **Prophet** (Taylor & Letham, 2018) | — | — | |
| **Seasonal Naïve** (Hyndman & Athanasopoulos, 2018) | — | — | |
| **STL-AR** (Talagala, 2021) | — | — | |
| **Theta** (Assimakopoulos & Nikolopoulos, 2000) | — | — | |
| | | **# Layers** | **# Cells** |
| **DeepAR** (Salinas et al., 2020a) | 1, 2, 4 | 1 | 20 |
| | | 2 | 40 |
| | | 4 | 80 |
| | | **# Filters** | **Kernel Sizes** |
| **MQ-CNN** (Wen et al., 2017) | 2, 4, 8 | 20 | 3, 3, 2 |
| | | 30 | 7, 3, 3 |
| | | 40 | 14, 7, 3 |
| **MQ-RNN** (Wen et al., 2017) | 2, 4, 8 | — | |
| | | **# Stacks** | **# Blocks** |
| **N-BEATS** (Oreshkin et al., 2019) | 1, 2, 4 | 4 | 5 |
| | | 30 | 1 |
| | | 30 | 2 |
| | | **Hidden Dim** | **# Layers** |
| **Simple Feedforward** | 1, 2, 4 | 30 | 1 |
| | | 40 | 2 |
| | | 80 | 3 |
| | | **Hidden Dim** | **# Heads** |
| **Temporal Fusion Transformer** (Lim et al., 2021) | 1, 2, 4 | 16 | 2 |
| | | 32 | 4 |
| | | 64 | 8 |

# A  MODELS

Table 2 provides an overview over all benchmark models along with hyperparameters considered. In addition to three hyperparameter settings per deep learning model (except for MQ-RNN), three different context lengths $l$ are considered. These represent multiples of the forecast horizon $\tau$ and are thus dataset-dependent. Eventually, this results in 9 model configurations per deep learning model (and 3 for MQ-RNN).

The implementation of all models are taken from GluonTS (Alexandrov et al., 2020). All deep learning models are implemented in MXNet (Chen et al., 2015). ARIMA, ETS, STL-AR, and Theta are available in GluonTS as well but forward computations to the R *forecast* package (Hyndman et al., 2021).

**Model Evaluation.** To measure the accuracy of forecasting models, we employ the *continuous ranked probability score* (CRPS) (Matheson & Winkler, 1976; Gneiting & Raftery, 2007). Given the quantile function $(F_t^{(i)})^{-1}$ of the forecast probability distribution in Eq. (1) for a time series $\boldsymbol{z}^{(i)}$

at time step $t$, the CRPS is defined as

$$\mathrm{CRPS}\left((F_t^{(i)})^{-1}, z_t^{(i)}\right) = \int_0^1 \Lambda_\alpha\left((F_t^{(i)})^{-1}, z_t^{(i)}\right) \, \mathrm{d}\alpha \tag{11}$$

where $\Lambda_\alpha$ is the quantile loss (or pinball loss) defined over a quantile level $0 < \alpha < 1$:

$$\Lambda_\alpha\left(F^{-1}, z\right) = \left(\alpha - \mathbb{I}[z < F^{-1}(\alpha)]\right)\left(z - F^{-1}(\alpha)\right) \tag{12}$$

Here, $\mathbb{I}$ is the indicator function. To compute an aggregated score for a model forecasting multiple time series and time steps, we compute a normalized CRPS (nCRPS):

$$\mathrm{nCRPS}\left(\left\{(F_t^{(i)})^{-1}\right\}_{i,t}, \mathcal{Z}\right) = \frac{\sum_{i,t} \mathrm{CRPS}\left((F_t^{(i)})^{-1}, z_t^{(i)}\right)}{\sum_{i,t} |z_t^{(i)}|} \tag{13}$$

We approximate the nCRPS using the 10-quantiles $\alpha \in \{0.1, 0.2, 0.3, \ldots, 0.9\}$. While we focus on nCRPS for the evaluations in this paper, the benchmark dataset that we release contains four additional forecast accuracy metrics (MAPE, sMAPE, NRMSE, ND).

## B  DATASETS

Our benchmark provides 44 datasets which were obtained from multiple sources. 16 datasets were obtained though GluonTS (Alexandrov et al., 2020), 24 datasets are taken from the Monash Time Series Forecasting Repository (Godahewa et al., 2021), and the remaining 4 datasets were originally published as part of Kaggle[2] forecasting competitions. Table 3 provides basic statistics about all datasets.

### B.1  DATASET DESCRIPTIONS

In the following, we provide brief descriptions of all datasets and provide their sources. We link the original source (if possible) along with any work which pre-processed the data (if applicable). The datasets "Corporación Favorita", "Restaurant", "Rossmann", and "Walmart" — which were obtained from Kaggle — are processed by ourselves.

**Australian Electricity Demand** (O'Hara-Wild et al., 2021; Godahewa et al., 2021) provides the 30-minute electricity demand of five Australian states (New South Wales, Queensland, South Australia, Tasmania, Victoria). Time series range from January 2002 to April 2015.

**Bitcoin** (Godahewa et al., 2021) provides over a dozen of potential influencers of the price of *Bitcoin*. Among others, these influencers are daily hash rate, block size, or mining difficulty. We exclude one time series from the original dataset as its absolute values are $\geq 10^{18}$. Data is provided from January 2009 to July 2021.

**CIF 2016** (Štěpnička & Burda, 2017; Godahewa et al., 2021) is the dataset from the "Computational Intelligence in Forecasting" competition in 2016. One third of the time series originate from the banking sector while the remaining two thirds are generated artificially. In the competition, the time series have two different prediction horizons, however, GluonTS forces us to adopt only a single horizon (for which we choose the most common one).

**COVID Deaths** (Dong et al., 2020; Godahewa et al., 2021) provides the daily COVID-19 death counts of various countries between January 22 and August 21 in the year 2020.

**Car Parts** (Hyndman, 2015; Godahewa et al., 2021) provides intermittent time series of car parts sold monthly by a US car company between January 1998 and April 2002. Periods in which no parts are sold have a value of zero.

**Corporación Favorita** (Favorita, 2017) contains the daily unit sales of over 4,000 items at 54 different stores of the supermarket company *Corporación Favorita* in Ecuador. Unit sales were manually set to 0 on days where there was no data available. Data ranges from January 2013 to August 2017. All covariates that are provided in the Kaggle competition are discarded.

---

[2]https://www.kaggle.com/

Table 3: Statistics of the benchmark datasets. Frequencies read as follows: MIN – minutely, H – hourly, D – daily, B – business daily, W – weekly, M – monthly, Q – quarterly, Y – yearly. Number of observations are calculated from the training data.

| | Freq. | Horizon | # Series | Avg. Length | # Observations |
|---|---|---|---|---|---|
| **Australian Electricity Demand** | 30 MIN | 48 | 5 | 231,005 | 1,155,024 |
| **Bitcoin** | D | 30 | 17 | 4,134 | 70,279 |
| **CIF 2016** | M | 12 | 72 | 87 | 6,244 |
| **COVID Deaths** | D | 30 | 227 | 182 | 41,314 |
| **Car Parts** | M | 12 | 2,504 | 39 | 97,656 |
| **Corporación Favorita** | D | 16 | 171,091 | 1,089 | 186,310,518 |
| **Dominick** | W | 8 | 115,163 | 157 | 18,137,043 |
| **Electricity** | H | 24 | 321 | 21,044 | 6,755,124 |
| **Exchange Rate** | B | 30 | 8 | 6,071 | 48,568 |
| **Fred-MD** | M | 12 | 107 | 716 | 76,612 |
| **Hospital** | M | 12 | 767 | 72 | 55,224 |
| **KDD 2018** | H | 48 | 270 | 10,850 | 2,929,404 |
| **London Smart Meters** | 30 MIN | 48 | 5,559 | 29,903 | 166,231,248 |
| **M1 Monthly** | M | 18 | 617 | 73 | 44,892 |
| **M1 Quarterly** | Q | 8 | 203 | 41 | 8,320 |
| **M1 Yearly** | Y | 6 | 181 | 19 | 3,429 |
| **M3 Monthly** | M | 18 | 1,428 | 99 | 141,858 |
| **M3 Other** | Q | 8 | 174 | 69 | 11,933 |
| **M3 Quarterly** | Q | 8 | 756 | 41 | 30,956 |
| **M3 Yearly** | Y | 6 | 645 | 22 | 14,449 |
| **M4 Daily** | D | 14 | 4,227 | 2,357 | 9,964,658 |
| **M4 Hourly** | H | 48 | 414 | 854 | 353,500 |
| **M4 Monthly** | M | 18 | 48,000 | 216 | 10,382,411 |
| **M4 Quarterly** | Q | 8 | 24,000 | 92 | 2,214,108 |
| **M4 Weekly** | W | 13 | 359 | 1,022 | 366,912 |
| **M4 Yearly** | Y | 6 | 22,974 | 31 | 707,265 |
| **M5** | D | 28 | 30,490 | 1,885 | 57,473,650 |
| **NN5** | D | 56 | 111 | 735 | 81,585 |
| **Pedestrian Count** | H | 48 | 66 | 47,412 | 3,129,178 |
| **Restaurant** | D | 39 | 823 | 321 | 263,817 |
| **Rideshare** | H | 48 | 2,304 | 493 | 1,135,872 |
| **Rossmann** | D | 48 | 1,115 | 864 | 963,689 |
| **San Francisco Traffic** | H | 48 | 862 | 17,496 | 15,081,552 |
| **Solar** | H | 24 | 137 | 7,009 | 960,233 |
| **Taxi** | 30 MIN | 24 | 1,214 | 1,488 | 1,806,432 |
| **Temperature Rain** | D | 30 | 32,053 | 695 | 22,276,835 |
| **Tourism Monthly** | M | 24 | 366 | 275 | 100,496 |
| **Tourism Quarterly** | Q | 8 | 427 | 92 | 39,128 |
| **Tourism Yearly** | Y | 4 | 518 | 21 | 10,685 |
| **Vehicle Trips** | D | 30 | 314 | 100 | 31,361 |
| **Walmart** | W | 39 | 2,934 | 102 | 298,509 |
| **Weather** | D | 30 | 3,010 | 14,266 | 42,941,700 |
| **Wiki** | D | 30 | 9,535 | 762 | 7,265,670 |
| **Wind Farms** | MIN | 60 | 313 | 513,442 | 160,707,330 |

**Dominick** (James M. Kilts Center, 2020; Godahewa et al., 2021) contains time series with the weekly profits of numerous stock keeping units (SKUs). The data is obtained from the grocery store company *Dominick's* over a period of seven years.

**Electricity** (Dua & Graff, 2017; Alexandrov et al., 2020) is comprised of the hourly electricity consumption (in kWh) of hundreds of households between January 2012 and June 2014.

**Exchange Rate** (Lai et al., 2018; Alexandrov et al., 2020) provides the daily exchange rates (on weekdays) between the US dollar and the currencies of eight countries (Australia, Great Britain, Canada, Switzerland, China, Japan, New Zealand, Singapore) in the period from 1990 to 2013.

**Fred-MD** (McCracken & Ng, 2016; Godahewa et al., 2021) contains monthly time series of macro-economic indicators (e.g. interest rates, employment rates, . . . ) from the Federal Reserve Bank since January 1959 (until September 2019).

**Hospital** (Hyndman, 2015; Godahewa et al., 2021) provides the monthly patient counts for various products that are related to medical problems between 2000 and 2007.

**KDD 2018** (Bekkerman et al., 2018; Godahewa et al., 2021) is the dataset used by the KDD Cup 2018. It contains hourly time series with air quality levels (represented by various chemical compounds) measured by stations in Beijing and London between January 2017 and April 2018.

**London Smart Meters** (Jean-Michael, 2019; Godahewa et al., 2021) was originally published on Kaggle and contains the half-hourly (electrical) energy consumption (in kWh) of thousands of households in the period from November 2011 to February 2014.

**M1** (Makridakis et al., 1982; Godahewa et al., 2021) datasets are taken from the M1 competition in 1982. Time series have varying start- and end dates and cover a wide semantic spectrum of domains (demographics, micro, macro, industry).

**M3** (Makridakis & Hibon, 2000; Alexandrov et al., 2020) datasets are obtained from the M3 competition. Across frequencies, time series were collected from six different domains: demographics, micro, macro, industry, finance, and other. For the time series with no specified frequency, we assume a quarterly frequency (as this is the default in GluonTS).

**M4** (Makridakis et al., 2020b; Alexandrov et al., 2020) datasets are taken from the prestigious M4 competition. The time series are collected from the same domains as in the M3 competition.

**M5** (Makridakis et al., 2020a; Alexandrov et al., 2020) is the dataset from the M5 competition and contains daily unit sales of 3,049 products across 10 *Walmart* stores located in California, Texas, and Wisconsin. While covariates are available (e.g. product prices or special events), none of the forecasting methods we consider makes use of them. Sales data is provided from January 2011 to April 2016.

**NN5** (Taieb et al., 2012; Godahewa et al., 2021) data was used in the NN5 competition run in 2008. The time series provide daily amounts of cash withdrawals at dozens of ATMs in the UK from March 1996 to May 1998.

**Pedestrian Count** (City of Melbourne, 2017; Godahewa et al., 2021) provides hourly pedestrian counts in Melbourne from May 2009 to May 2020. The counts are captured by sensors scattered in the city.

**Restaurant** (Holdings, 2017) provides the number of daily visitors of hundreds of restaurants in Japan between January 2016 and April 2017, as tracked by the *AirREGI* system. As dates for which restaurants are closed do not provide visitor numbers, we impute zeros for these missing values. The Kaggle competition where this dataset is obtained from also provides data about reservations but this data remains unused for our purposes.

**Rideshare** (Godahewa et al., 2021) is comprised of hourly time series representing various attributes related to services offered by *Uber* in *Lyft* in New York. Time series include e.g. minimum, maximum, and average price, or the maximum distance traveled — grouped by starting location and provider. Data is available for the period of a month, starting in late November 2018.

**Rossmann** (Rossmann, 2015) provides daily sales counts at hundreds of different stores of the *Rossmann* drug store chain. Just like for the M5 dataset, covariates are available (e.g. distance to competition, special events, or discounts) but not used by any method. Data is available from January 2013 until August 2015.

**San Francisco Traffic** (Caltrans, 2020; Godahewa et al., 2021) contains hourly occupancy rates of freeways in the San Francisco Bay area. Data is available for two full years, starting from January 1, 2015.

**Solar** (Zhang, 2006; Alexandrov et al., 2020) is comprised of the hourly power consumption of dozens of photovoltaic power stations in the state of Alabama in the year 2006.

**Taxi** (NYC Taxi and Limousine Commission, 2015; Salinas et al., 2019; Alexandrov et al., 2020) contains the number of taxi rides in hundreds of locations around New York in 30 minute windows.

Training data contains data from January 2015 while test data contains data from January in the subsequent year.

**Temperature Rain** (Godahewa et al., 2021) provides the daily temperature observations and rain forecasts gather from 422 weather stations across Australia. Data ranges from May 2015 through April 2017.

**Tourism** (Athanasopoulos et al., 2011; Godahewa et al., 2021) datasets come from a Kaggle forecasting competition where time series were supplied by tourism bodies from Australia, Hong Kong, and New Zealand as well as academics. Monthly data ranges from 1979 through 2007, quarterly data from from 1975 through 2007, and yearly data from 1960 through 2008.

**Vehicle Trips** (Flowers & Fischer-Baum, 2015; Godahewa et al., 2021) contains the daily number of trips served by hundreds of for-hire vehicle (FHV) companies (such as *Uber*). Data ranges from January to September 2015.

**Walmart** (Walmart, 2014) contains data about *Walmart* store sales similarly to the M5 competition. Data is provided as the weekly sales (in USD) across 45 stores and 81 departments in different regions and ranges from February 2010 to November 2012. Just like for many of the other Kaggle competitions, covariates are available, yet unused.

**Weather** (Sparks, 2021; Godahewa et al., 2021) is comprised of daily data collected from hundreds of weather stations in Australia. Collected data is the amount of rain, the minimum and maximum temperature, and the amount of solar radiation.

**Wiki** (Gasthaus et al., 2019; Alexandrov et al., 2020) similarly provides daily pages views for several thousand Wikipedia pages in the period from January 2012 to July 2014.

**Wind Farms** (AEMO, 2020; Godahewa et al., 2021) contains time series with the minutely power production of hundreds of wind farms in Australia. Data is available for a period of one year, starting in August 2019.

## B.2 DATA PREPARATION

For all of the datasets in Table 3 except for "Taxi", we perform similar preprocessing. Let $\mathcal{Z} = \{\boldsymbol{z}_{1:T_i}^{(i)}\}_{i=1}^{K}$ be the set of $K$ univariate time series of lengths $T_i$. Then, we run the following preprocessing steps:

1. Remove all time series which are constant to always be able to compute the MASE metric.

$$\mathcal{Z}_1 = \{\boldsymbol{z}^{(i)} \in \mathcal{Z} \mid \neg \exists c \in \mathbb{R} : \boldsymbol{z}^{(i)} = c \cdot \mathbf{1}_{T_i}\} \tag{14}$$

2. Remove all time series which are too short. We exclude all time series of length $T \leq (p+1)\tau$ where $\tau$ is the prediction horizon and $p$ is a dataset-specific integer which governs over how many prediction lengths we want to perform testing[3]. We are using $p = 1$ for all datasets but "Electricity" ($p = 7$), "Exchange Rate" ($p = 3$), "Solar" ($p = 4$), "Traffic" ($p = 7$) and "Wiki" ($p = 3$).

$$\mathcal{Z}_2 = \{\boldsymbol{z}^{(i)} \in \mathcal{Z}_1 \mid T_i > (p+1)\tau\} \tag{15}$$

3. Split time series into training, validation and test data. For each time series, we split off the prediction horizon $p - 1$ times to obtain time series for the test data. Then, we split off the prediction horizon another time to obtain a validation time series per training time series. Eventually, we construct the following sets:

$$\mathcal{Z}_{\text{test}} = \bigcup \left\{ \{\boldsymbol{z}_{1:T_i}^{(i)}, \ldots, \boldsymbol{z}_{1:(T_i - (p-1)\tau)}^{(i)}\} \mid \boldsymbol{z}^{(i)} \in \mathcal{Z}_2 \right\} \tag{16}$$

$$\mathcal{Z}_{\text{val}} = \left\{ \boldsymbol{z}_{1:(T_i - p\tau)}^{(i)} \mid \boldsymbol{z}^{(i)} \in \mathcal{Z}_2 \right\} \tag{17}$$

$$\mathcal{Z}_{\text{train}} = \left\{ \boldsymbol{z}_{1:(T_i - (p+1)\tau)}^{(i)} \mid \boldsymbol{z}^{(i)} \in \mathcal{Z}_2 \right\} \tag{18}$$

---

[3]Assume $p = 2$ and a time series $\boldsymbol{z} \in \mathbb{R}^T$ from a dataset with prediction horizon $\tau$. Then, we perform testing on time series $\boldsymbol{z}^{(1)} = \boldsymbol{z}_{1:T}$ and $\boldsymbol{z}^{(2)} = \boldsymbol{z}_{1:(T-\tau)}$ while $\boldsymbol{z}^{(3)} = \boldsymbol{z}_{1:(T-2\tau)}$ is being trained on.

For the "Taxi" dataset, we need to slightly augment the preprocessing to allow for discontinuities in the available time series. Essentially, each time series $z \in \mathcal{Z}$ consists of an old part $z_{\text{old}}$ and a new part $z_{\text{new}}$ (data throughout January of two successive years). We then obtain the training data from $z_{\text{old}}$ and construct the validation data by cutting the prediction horizon from $z_{\text{old}}$. The test dataset is constructed from $z_{\text{new}}$ as in bullet point 3 with $p = 56$.

## C   TRAINING DETAILS

This section describes in detail how the deep learning models from Table 2 are trained on the datasets outlined in the previous section. As outlined in Section 3.2, parametric local methods directly estimate their parameters at training time. While deep learning models are trained on $\mathcal{Z}_{\text{train}}$ such that models at different checkpoints can be compared using $\mathcal{Z}_{\text{val}}$, parameters of parametric local methods are estimated using $\mathcal{Z}_{\text{val}}$.

### C.1   TRAINING TIME

For all deep learning models, we run training for a predefined duration depending on the dataset size. For $n$ total observations in the training data, we run training for $d$ hours where

$$d = 2^\eta \qquad \text{with} \qquad \eta = \min\left\{\max\left\{\left\lfloor \log_{10}\left(\frac{n}{10000}\right)\right\rfloor, 0\right\}, 3\right\}. \tag{19}$$

Training therefore always runs between one and eight hours (refer to Table 3 for training times on individual datasets). Even on small datasets, we train for an hour to ensure that the trained model receives sufficiently many gradient updates.

### C.2   PROBABILISTIC FORECASTS

Forecasts of all methods are required to be probabilistic to store the 10-quantiles $q_{0.1}, q_{0.2}, \ldots, q_{0.9}$. While some methods forecast the quantiles directly (e.g. MQ-CNN, TFT), other methods provide samples of the output distribution (e.g. DeepAR) or point forecasts (e.g. N-BEATS). For sampling-based models, we simply compute the empirical 10-quantiles. For the point forecasts, we interpret forecasts as Dirac distributions centered around the forecasted value $y$. Thus, each quantile can be set to the value $y$.

### C.3   LEARNING RATE SCHEDULING

All deep learning models are trained using the Adam optimizer (Kingma & Ba, 2014) with an initial learning rate of $\rho = 10^{-3}$. During training, we decrease the learning rate three times by a factor of $\lambda = \frac{1}{2}$ using a linear schedule. When training for a duration $d$, we decrease the learning rate after durations $\mathcal{D} = \{\frac{d}{4}, \frac{d}{2}, \frac{3d}{4}\}$. This results in a final learning rate of $\lambda^3 \rho = 1.25 \cdot 10^{-4}$. The decay is always applied after the first batch exceeding one of the durations in $\mathcal{D}$.

### C.4   VALIDATION AND TESTING

During training, we save the model and compute its loss $\mathcal{L}$ as well as the nCRPS $\mathcal{Q}$ on the validation data for a total of 11 times. For a model being trained for a duration $t$, we perform these computations after durations $\mathcal{D} = \mathcal{H} \cup \{k\frac{t}{9} \mid k \in \{1, \ldots, 9\}\}$ with intervals $\mathcal{I} = \{t \cdot 3^{-k} \mid k \in \{0, \ldots, 4\}\}$. For each $d \in \mathcal{D}$, we run validation on $\mathcal{Z}_{\text{val}}$ after the first batch where the total training time exceeds $d$ and compute the losses $\mathcal{L}_d$ and $\mathcal{Q}_d$.

Note that we do not add the time taken to run validation to the training time to evaluate whether it exceeds $d$. We argue that validation can be run swiftly. If necessary, it could also be run on a subset of all available time series for further speedup.

For testing, we choose 5 of the 11 models that we saved during training. For each $d \in \mathcal{I}$, we choose the model with the lowest nCRPS that was encountered up to $d$. Note that this may result in choosing the same model multiple times if continued training did not yield any improvements on the validation data.

## C.5 INFRASTRUCTURE AND TRAINING STATISTICS

All training jobs were scheduled on AWS Sagemaker[4] using CPU-only `ml.c5.2xlarge` instances (8 CPUs, 16 GiB memory). In few cases, we ran jobs on other instance types:

- `ml.m5.2xlarge` instances (8 CPUs, 32 GiB memory) to counteract out-of-memory-issues. Used for N-BEATS on "Corporación Favorita" and "Weather", DeepAR on "Corporación Favorita", and Prophet on "Wind Farms".

- `ml.m5.4xlarge` instances (16 CPUs, 64 GiB memory) to allow for even more memory-intensive models. Used for N-BEATS on "London Smart Meters".

- `ml.c5.18xlarge` instances (72 CPUs, 144 GiB memory) to speed up predictions by parallelization. Used for ARIMA on "Taxi".

In total, we ran 4,708 training jobs. Including validation and testing, this amounted to roughly 684 days of total training time ($\sim$3 hours and 30 minutes per training job). Parallelized over 200 instances, actual training time for our benchmark was roughly 4 days and amounts to approximately $7,300 of compute costs on AWS.

## D SURROGATE MODEL

In the following, we discuss the choice of the surrogate model. First, we outline why a parametric surrogate model is desirable. Then, we describe how model configurations are vectorized to be used by parametric surrogate models. Afterwards, we list the ranking loss with linear discounting that we mention in Section 4.2 and eventually compare the ranking performance of different surrogate models.

### D.1 CHOICE OF SURROGATE MODEL

Intuitively, a nonparametric surrogate model might appear to be optimal for ranking models. However, a parametric surrogate has several advantages. First, it can interpolate between neighboring configurations, which is essential when the configurations differ between different datasets or when the objective is noisy. Second, a parametric model allows for suggesting new configurations that e.g. maximize capacity under a given constraint. Third, a parametric surrogate model $\tilde{f}_\theta$ may also be conditioned on the dataset by taking as input dataset features (such as number of time series or observations in the task at hand) by concatenating features to the input vector of $\tilde{f}_\theta$ — although we do not consider this possibility here.

### D.2 INPUT VECTORIZATION

When using XGBoost or an MLP as surrogate model for predicting model performance, models must be represented as feature vectors. For this, we proceed as follows to encode model configurations:

- The model type (i.e. ARIMA, DeepAR, ...) is encoded via a one-hot encoding, yielding a 13-dimensional vector.

- Every model hyperparameter (across models) defines a new real-valued feature. Hyperparameters that are shared among deep learning models (context length multiple, training time, learning rate[5]) are re-used across models. This results in 15 additional features which are all standardized to have zero mean and unit variance.

Inputs to the MLP are, thus, 28-dimensional. Features that are missing (e.g. all classical models do not provide a context length multiple) are imputed by the features' means (resulting in zeros due to the standardization).

---

[4]https://aws.amazon.com/sagemaker/
[5]Although never altered, this provides a potential hyperparameter.

## D.3 Ranking Loss with Linear Discounting

Section 4.2 outlines that the MLP surrogate model $\tilde{\boldsymbol{f}}_\theta$ is trained via listwise ranking using linear discounting. For training on the offline evaluations $\mathcal{D}$, $\tilde{\boldsymbol{f}}_\theta$ minimizes the loss $\mathcal{L}$ via SGD. As ranking is performed for each objective $k \in \{1, \ldots, m\}$ and task $j \in \{1, \ldots, T\}$ independently, $\mathcal{L}$ can be written as follows:

$$\mathcal{L}(\tilde{\boldsymbol{f}}_\theta) = \sum_{k=1}^{m} \sum_{j=1}^{T} \mathcal{L}_{kj}(\tilde{f}_{\theta;k}) \tag{20}$$

where $\tilde{f}_{\theta;k}$ yields the surrogate model's outputs for objective $k$ and $\mathcal{L}_{kj}(\tilde{f}_{\theta;k})$ is defined as follows (where we ignore indices $k$ and $j$ in the formula for notational convenience):

$$\mathcal{L}_{kj}(\tilde{f}_{\theta;k}) = \sum_{i=1}^{N} \frac{1}{i} \left( \tilde{f}_\theta \left( x_{r(i)} \right) + \log \sum_{l=i}^{N} \exp \left( -\tilde{f}_\theta \left( x_{r(i)} \right) \right) \right) \tag{21}$$

Here, $N$ provides the number of available evaluations for task $j$ and $r(i)$ defines the index of the configuration $x$ that is ranked at position $i$ with respect to objective $k$ among all configurations for task $j$. The ranks of the configurations $x_1, \ldots, x_N$ with respect to an objective $k$ can be computed from the corresponding evaluations $y_{1;k}, \ldots, y_{N;k}$. Note that rank $i = 1$ provides the index for the "best" configuration and $\tilde{f}_\theta$ outputs a smaller value for configurations that have a smaller rank (i.e. are "better").

## D.4 Comparison of Different Models

The following compares the MLP surrogate trained via listwise ranking introduced in Section 4 to other possible choices for the surrogate model. Much like the approach described in Section 4.3, we evaluate surrogate models by performing LOOCV: surrogates are evaluated on each of the 44 benchmark datasets one after the other, being trained on the remaining 43 datasets. Surrogates are evaluated with respect to different ranking metrics that give an overview of the models' ability to *rank* the nCRPS values of model configurations on different datasets. As metrics for measuring the ranking performance on a single dataset, we consider the following which yield metrics between 0 and 1:

- **Mean Reciprocal Rank (MRR)** (Liu, 2009): Computed as one divided by the predicted rank of the model with the lowest true nCRPS.

- **Precision@k** (Liu, 2009): The fraction of the $k$ models with the lowest true nCRPS captured in the top $k$ predictions.

- **Normalized Discounted Cumulative Gain (NDCG)** (Liu, 2009): A measure for the overall ranking performance. It first computes the discounted cumulative gain (DCG) by summing the discounted relevance scores $1, 0.9, \ldots, 0.1$ of the 10 models with the lowest true nCRPS scores — more specifically, a relevance score $\pi(i)$ is discounted by $\log_2 k + 1$ where $k$ is the predicted rank of the model with true rank $i$. Then, it normalizes the DCG by using the ideal DCG (iDCG) to obtain the NDCG.

As a comparison to the ranking MLP surrogate with linear discount (**MLP Ranking + Discounting**), we first consider a random surrogate which predicts nCRPS by sampling from $\mathcal{U}(0,1)$ as a baseline (**Random**). Then, we use a nonparametric model which predicts the nCRPS as the average among all training datasets (**Nonparametric Regression**) and one that predicts the average rank of the nCRPS across training datasets (**Nonparametric Ranking**). Additionally, we consider XGBoost surrogates which are trained via regression (**XGBoost Regression**) and pairwise ranking[6] (**XGBoost Ranking**), respectively. Lastly, we consider an MLP with the same architecture but trained via regression instead of listwise ranking (**MLP Regression**) and one without any discounting (**MLP Ranking**).

Table 4 clearly shows that the MLP with listwise ranking and linear discounting markedly outperforms all other choices for the surrogate model. Especially, it is much more capable of identifying

---

[6]XGBoost does not provide a working version of listwise ranking.

Table 4: Comparison of different surrogate models with respect to their ability to rank model configurations according to their nCRPS. Ranking metrics are averaged across all benchmark datasets and multiplied by 100. Metrics for non-deterministic surrogates (random and MLP) are further averaged by running the entire evaluation over five random seeds. Best values for each metric are displayed in bold.

| | MRR | NDCG | Precision@5 | Precision@10 | Precision@20 |
|---|---|---|---|---|---|
| **Random** | 2.17 | 25.36 | 2.45 | 4.68 | 8.27 |
| **Nonparametric Regression** | 6.88 | 35.80 | 13.18 | 20.91 | 32.73 |
| **Nonparametric Ranking** | 6.78 | 35.87 | 10.00 | 20.91 | 33.07 |
| **XGBoost Regression** | 5.68 | 35.34 | 13.18 | 22.05 | 32.61 |
| **XGBoost Ranking** | 6.64 | 35.63 | 12.27 | 21.59 | 32.84 |
| **MLP Regression** | 3.86 | 32.06 | 7.73 | 13.73 | 25.23 |
| **MLP Ranking** | 5.00 | 34.59 | 10.27 | 21.00 | 33.34 |
| **MLP Ranking + Discounting** | **16.41** | **43.74** | **23.82** | **28.32** | **34.39** |

the top configurations, stressed by the very high values of MRR and Precision@5 compared to the other surrogate models. This can likely be attributed to the linear discounting that encourages the model to focus on ranking the top configurations correctly. Thus, the loss formulation is more stable in the presence of outliers.

# E   PARETOSELECT PSEUDO CODE

---

**Algorithm 1:** Overview of PARETOSELECT

---

**Data:** Time series models $\mathcal{X}$, offline evaluations $\mathcal{D}$, number of default models $n$
**Result:** A set $\{x_1, \ldots, x_n\} \subset \mathcal{X}$ of model defaults

**function** non_dominated_sort($\mathcal{X}$, $\tilde{\boldsymbol{f}}_\theta$):
    $\Lambda = []$
    **while** $\mathcal{X} \neq \emptyset$ **do**
        $\mathcal{P}_{\mathcal{X},\tilde{\boldsymbol{f}}_\theta} = \{x \in \mathcal{X} \mid \neg \exists x' \in \mathcal{X} : x' \prec_{\boldsymbol{f}} x\}$     // Compute the Pareto front
        $\mathcal{X} = \mathcal{X} \setminus \mathcal{P}_{\mathcal{X},\tilde{\boldsymbol{f}}_\theta}$     // Remove Pareto front
        extend($\Lambda$, compute_epsilon_net($\mathcal{P}_{\mathcal{X},\tilde{\boldsymbol{f}}_\theta}$, $\tilde{\boldsymbol{f}}_\theta$))
    **return** $\Lambda$

**function** compute_epsilon_net($\mathcal{X}$, $\tilde{\boldsymbol{f}}_\theta$):
    $\lambda = [\text{pop}(\mathcal{X})]$     // Choose and remove a random element
    **while** $\mathcal{X} \neq \emptyset$ **do**
        $D = \{\}$     // Mapping from remaining elements to minimum distances
        **for** $x \in \mathcal{X}$ **do**     // Compute minimum distance to all chosen elements
            $D[x] = \min_{x' \in \lambda} \|\tilde{\boldsymbol{f}}_\theta(x) - \tilde{\boldsymbol{f}}_\theta(x')\|_2$
        // Append the element furthest away from the current selection
        append($\lambda$, pop($\mathcal{X}$, $\text{argmax}_{x \in \mathcal{X}} D[x]$))
    **return** $\lambda$

$\tilde{\boldsymbol{f}}_\theta = $ train_surrogate_model($\mathcal{D}$)
// Multi-objective sorting according to predictions
$x_1, \ldots, x_{|X|} = $ non_dominated_sort($\mathcal{X}$, $\tilde{\boldsymbol{f}}_\theta$)
**return** $x_1, \ldots, x_{\min(n,|X|)}$

---

# F    ANALYSIS OF DATASETS CHARACTERISTICS

In this section, we try to answer the following question: are there some simple dataset characteristics that can help to know which model is going to perform best? In particular, we analyze in details the results of Section 3.5 where we showed that, on some datasets, (1) statistical methods outperformed deep learning methods and that (2) the performance of statistical and deep learning methods could not be distinguished.

We start by analyzing more in depth the four out 44 datasets where deep learning models do not outperform classical methods — Figure 2 showed that the best deep learning method outperforms the best classical methods on all but four of our benchmark datasets. On these four datasets ("KDD2018", "M3 Yearly", "M4 Quarterly", "Taxi"), the summary statistics differ wildly as can be seen from Table 3 both in terms of the number of observations, the average length, and other characteristics. Further, the classical method that outperforms the best deep learning method is different for each of these datasets ("KDD2018": NPTS, "M3 Yearly": Theta, "M4 Quarterly": ETS, "Taxi": STL-AR). Notably, this does not include the local method that performs best on average (i.e. ARIMA, see Table 1).

Figure 1 showed that for 14 out of the 44 benchmark datasets, the performance of classical methods is indistinguishable from the performance of deep learning methods. To further study these datasets, we plot in Figure 7 the correlation matrix of all $k$ benchmark methods' nCRPS ranks across the datasets (using only the default configuration for the deep learning models). Notably, for twelve datasets ("Exchange Rate" through "M1 Yearly" on the y-axis), we observe a very low correlation with a large number of datasets. On these datasets, all models perform similarly due to strong stochasticity (e.g. "Exchange Rate", "Bitcoin", "M1 Quarterly", ...) or seasonality (e.g. "Tourism", "Ride share", ...) and predicting their ranking is therefore hard. Notably, 11 of these datasets overlap with the 14 datasets where classical methods are indistinguishable from deep learning methods.

In line with these observations, we found that adding simple dataset grouping features such as domain type (electricity, retail, finance) as an input for a surrogate model predicting the performance of forecasting methods does not have much effect. Our conclusion is that the ranks of the best performing models seem to be relatively hard to identify only from simple dataset grouping rules. Nonetheless, we hope that by releasing the benchmark data, we help future research to identify dominant methods from dataset characteristics.

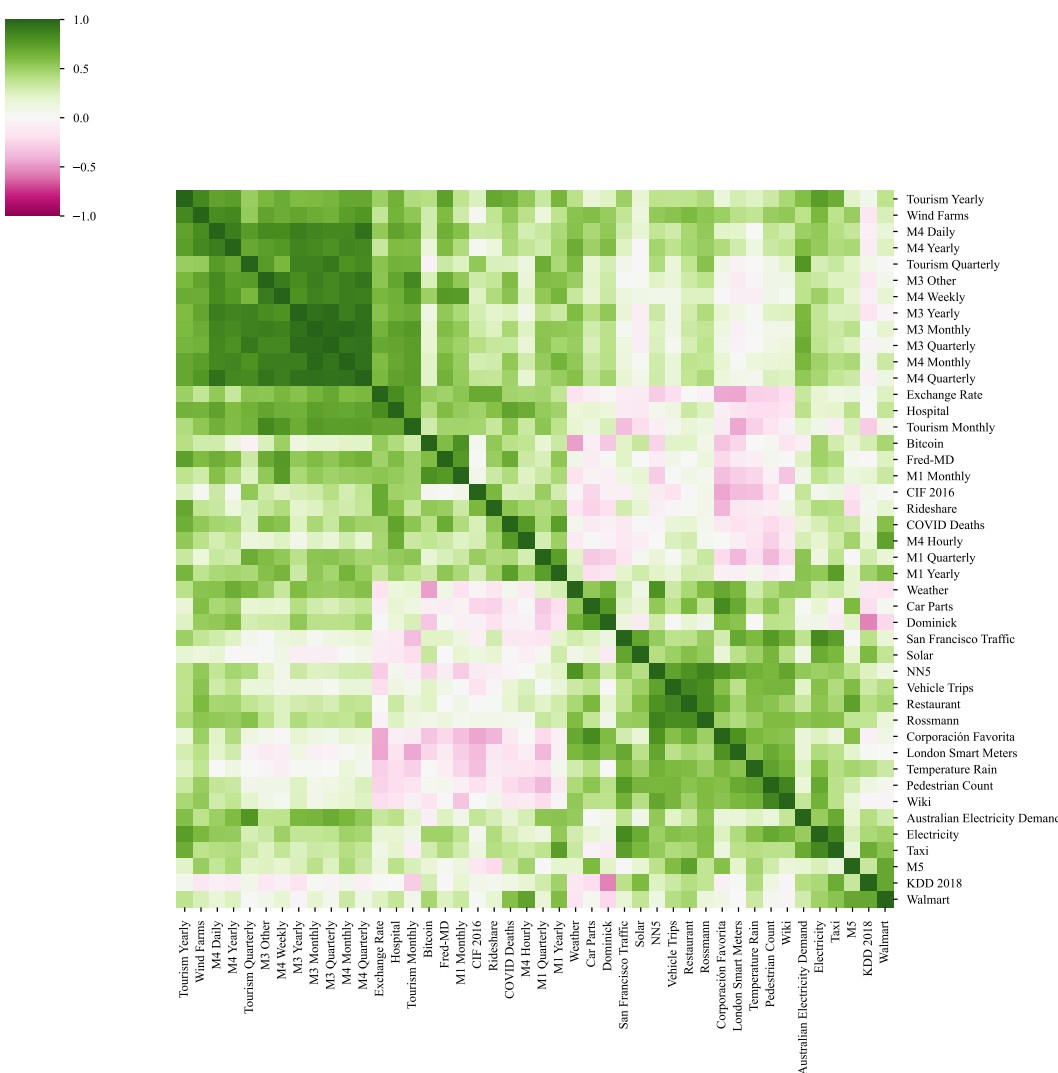

Figure 7: Correlation matrix of all benchmark methods' nCRPS ranks across all benchmark datasets. For the computation of the ranks, all classical methods and the default configuration of deep learning models are considered.

