# OpenReview forum: "Multi-Objective Model Selection for Time Series Forecasting"
_ICLR.cc/2022/Conference — ICLR 2022 Submitted_

### Official Review · Reviewer_rQb3 · 2021-10-24

**Correctness:** 3
**Technical Novelty And Significance:** 2
**Empirical Novelty And Significance:** 3
**Recommendation:** 5
**Confidence:** 4

**Main Review:**

**Strengths**:

1. Thorough empirical studies on both classical and state-of-the-art forecasting models on different datasets provide insights to researchers and practitioners on choosing appropriate models. The result itself is valuable: e.g., comparison between classical and NN-based methods, global and local models, etc.

2. Learning models in a multi-objective setting is a common problem in many applications. Constructing the model configuration set based on the Pareto front is a natural solution and is proven to be effective over other baselines.

**Weakness**:

1. The overall structure of this paper can be improved. For example, the paper title is ''Multi-Objective Model Selection ...'', but the authors start discussing this topic in the second half of this paper with limited context. This can easily cause misunderstanding to readers what is the major focus of this work.

2. Although the empirical study is an important component, there are still too many experimental details in the paper. It is better to elaborate on some employed methodologies. For example, instead of a plain citation, the authors could add brief introductions on the hypervolume error and $\epsilon$-net.

3. Since the property of each dataset varies a lot, the authors should also consider providing insights on which type of model is more suitable to datasets with certain properties, while the current paper aggregate the data information too much, and just provide general results on all datasets.

**Other Questions**:

1. Section 4.2, what is the reason that the parameters of MLP are not tuned?

2. What are the layers in figure 3? Are these layers of MLP or others?

3. For models in the Pareto front, how to finally trade off each objective in the experiments?

4. Section 4.4, why are non-uniform weights not helpful in ensemble? Consider an extreme case where the model is just a naive average, then it is reasonable to assign less weight to this model compared with others. If the set of model candidates has already been very good, then the weight distribution should be close to uniform in the first place. Authors could add more clarifications on this matter.

**Summary Of The Paper:**

This paper provides comprehensive evaluations of various time series forecasting models on 44 heterogeneous and public datasets. The obtained results, which are evaluated in multiple metrics, can serve as guidance for future researchers on choosing appropriate time series models for a given task. In addition, the authors have also studied how to obtain (or search) a set of default models that can provide good performance under single and multiple objective scenarios. For multi-objective optimization, a method called ParetoSelect is proposed to find the set of models that are in the Pareto front. Results have shown that ParetoSelect has outperformed other model selection methods.

**Summary Of The Review:**

This paper has conducted a thorough empirical study on forecasting models, but it still needs some major modifications on paper structures and clarifications on methodologies before being accepted.

## Update after Rebuttal

We thank the authors for their detailed response in addressing questions and updating their manuscripts. The analysis of dataset characteristics in Appendix F is interesting but it does not appear to provide useful information that can serve as an insight in the future: this is fine on my side since it is not easy to discover patterns on massive datasets.
Although the authors have also addressed the minor questions, we still think the structure of the paper can be further improved to make the paper more readable. The authors need some non-trivial modifications before it can be accepted. We wish the authors good luck in future revisions.

---

> ### Author Response · Authors · 2021-11-22
> **Response to Reviewer rQb3**
>
> We thank the reviewer for their comments. We address individual concerns below.
>
> **_“Since the property of each dataset varies a lot, the authors should also consider providing
> insights on which type of model is more suitable to datasets with certain properties, while the
> current paper aggregate the data information too much, and just provide general results on all
> datasets.”_**
>
> As indicated to reviewer FZ2f, we could not identify any patterns in the datasets that would be
> indicative of which method performs best. For instance, for the “outlier” datasets that are
> mentioned (“KDD2018”, “M3 Yearly”, “M4 Quarterly”, “Taxi”), the summary statistics differ wildly as
> can be deduced from Table 3 in the appendix. Further, the local model that outperforms the best
> deep learning method is different for each of these datasets (“KDD2018”: NPTS, “M3 Yearly”: Theta,
> “M4 Quarterly”: ETS, “Taxi”: STL-AR). Notably, this does not even include the local method that
> performs best on average (i.e. ARIMA). We added this analysis to Appendix F. We hope releasing the
> benchmark data may help future research to identify the best method from dataset characteristics.
>
> **_“Section 4.2, what is the reason that the parameters of MLP are not tuned?”_**
>
> We did not observe much gain by tuning the MLP hyper-parameters and therefore resulted to a default
> value. The biggest impacting factor was the use of a ranking loss with discounting (performance of
> different predictive surrogates is given in Table 4 of the appendix).
>
> **_“What are the layers in figure 3? Are these layers of MLP or others?”_**
>
> Layers in Figure 3 refer to “layers” of Pareto fronts for the non-dominated sorting algorithm
> (NDSA): the first layer is the Pareto front of set of points considered. When this Pareto front is
> removed, the new Pareto front constitutes the second layer, etc. This terminology is adapted from
> the literature and has been used in previous descriptions of NDSA (e.g. Emmerich et al., 2018). We
> believe that this description is covered by the last sentence on page 7.
>
> **_“For models in the Pareto front, how to finally trade off each objective in the experiments?”_**
>
> Since there are multiple objectives, there is no a priori final trade-offs between the objectives,
> the goal of multi-objective optimization is therefore to find the _set_ of models that are optimal.
> That being said, one way to select a final model is to pick the most accurate model under a given
> user latency constrain or to scalarize all the objectives into a single one (for instance, if all
> objectives can be expressed in dollars, one can pick a single model minimizing the dollar cost).
>
> **_“Section 4.4, why are non-uniform weights not helpful in ensemble? Consider an extreme case
> where the model is just a naive average, then it is reasonable to assign less weight to this model
> compared with others. If the set of model candidates has already been very good, then the weight
> distribution should be close to uniform in the first place. Authors could add more clarifications
> on this matter.”_**
>
> First, we want to refer to prior work on ensembling time series forecasts. Primarily, we want to
> refer to [1, Section 2.6.4]: “the simple average [...] has been shown to be a surprisingly robust
> combined forecast in the case of point forecasting [...]. [It] often outperforms more complicated
> point aggregations schemes such as weighted combinations”. Petropolous et al. refer to [2] and [3]
> for the latter statement.
>
> Nonetheless, we considered two settings other than simple forecast averaging in our experiments.
> (1) We used the nCRPS $Q_i$ of model $i$ and weighted each model by $1 / Q_i$ (and normalized
> weights to sum to 1). (2) We ranked the nCRPS values using ranks $\{0, ..., n-1\}$ (for ensemble
> size $n$) and derived weights by running a softmax over the ranks. In both cases, models with a
> lower nCRPS are assigned a (much) higher weight. We observed that — both when using the predicted
> nCRPS by our surrogate model and when accessing the true nCRPS — simple averaging of the forecasts
> performs better on average.
>
> We did not further specify these weighting schemes in the paper due to their relative complexity
> and empirical ineffectiveness. The last sentence of the second paragraph in Section 4.4 already
> touches upon our experimental findings that simple averaging is most effective. We rephrased this
> sentence and added another one to include a pointer to [1].
>
> ---
>
> [1] Petropoulos, Fotios et al. “Forecasting: Theory and Practice”. In: arXiv preprint
> arXiv:2012.03854 (2021).
>
> [2] Smith, J., Wallis, K. F. “A simple explanation of the forecast combination puzzle”. In: Oxford
> Bulletin of Economics and Statistics (2009).
>
> [3] Soule, David, Grushka-Cockayne, Yael, and Merrick, Jason RW. "A Heuristic for Combining
> Correlated Experts When There is Little Data." In: SSRN (2021).

---

### Official Review · Reviewer_FZ2f · 2021-11-03

**Correctness:** 3
**Technical Novelty And Significance:** 2
**Empirical Novelty And Significance:** 3
**Recommendation:** 5
**Confidence:** 4

**Main Review:**

The paper addresses the important question of a large-scale assessment of distributional time series forecasting algorithms, across a wide range of benchmark datasets, comparing both local methods (classical statistical techniques) with global ones (mostly deep learning approaches). Such a large-scale comparison is missing from the literature, and represents a gap in our current understanding of which method best apply to which context. The paper restricts its focus to univariate assessment in discrete time without covariates, which is a reasonable starting point.

In addition, the paper introduces a method to select hyperparameters and ensembles based on approximation of the Pareto front in a multi-objective setting. This approach is interesting and to the best of my knowledge is novel in a time series context.

The paper is reasonably well written and easy to follow, although the mathematical notation and writing style could be made more precise, as a few points are listed below. The authors are to be commended for providing full source code to reproduce experiments, and a cursory look suggests that the code is well written and easy to understand.

The main concern with the paper is that its focus is unclear as it tries to do both too much and too little:

- If viewed as a contribution to the large-scale benchmarking of time series forecasting algorithms, it offers a starting point but falls far short in offering substantive analysis of the presented results, e.g. to convey strong conclusions as to the suitability of particular methods for particular datasets, e.g. regarding time series properties such as variability, intermittency, seasonalities, etc., and possibly grouping benchmark datasets by broad area of study (e.g. retail, economics, energy, ...).
- If viewed as contributing the `ParetoSelect` algorithm, then this should come much earlier in the paper (in its current position towards the end, it feels like an afterthought), and deeper analysis of the results should be presented, along with a theoretical analysis if possible.

As written, the paper attempts to cover both objectives, but falls short of offering strong guidance or conclusions. It is therefore difficult to appreciate the paper's takeaways for the ICLR audience, and as such, it would require more work before recommending acceptance to the conference.

Detailed comments:
- Typographic remark: please put table captions above the table, and figure captions below the figure. (https://tex.stackexchange.com/questions/3243/why-should-a-table-caption-be-placed-above-the-table)
- p. 3 "three different context lengths that governs" ==> govern
- Section 3.4: since the forecast horizons and history length differ widely, isn't it better to use a *relative* latency measure compared to some simple benchmark (e.g. seasonal naïve)? Otherwise, the latency results are dominated by the datasets with the longest time series, regardless of the model performance per time step.
- Section 3.5: The results reported in Table 1 include rank information across an unknown set of models: that's obviously more than 7+6=13 introduced in Section 3.2, and seems to include some form of ensembling. However, how the ensembles are constructed has not yet been introduced by that point in the paper.
- p.5 [Comparison of classical & deep learning methods]: the paper should explain how the test statistic is constructed exactly. Does the test include only the raw deep learning methods, or it includes ensembles as well? Can anything be learned from the datasets for which local methods outperform global methods apart from the number of examples in the dataset?
- p. 5: "only a few thousands observations seem sufficient to outperform the classical models considered": in Figure 1, there are datasets with >1e6 observations where local methods are indistinguishable from global methods, this is way more than "a few thousands".
- p. 6 [top] it seems that $\mathcal{Q}_\mathrm{deep}$ and $\mathcal{Q}_\mathrm{class}$ represent different model results (best models of each class, respectively) than those given a few paragraphs above (set of all models within each class). It would help readability to have a more precise and coherent notation.
- Section 4.1: it is not clear if the $M$ different tasks represent different datasets or different objectives. And why does eq. (5) evaluates on the $M$ related tasks only, but not on the main task? The notation should be tightened up.
- Eq (12): it's not clear how sort is defined for vectors, and does not easily relate to the numbering of datapoints shown in Figure 3.
- The bibliographic entries should be checked for completeness. For instance Pfisterer et al. (2021) and Winkelmolen et al. (2020) are incomplete.
- p. 20: "we set each quantile to the forecasted value" ==> do you mean that $q_{0.5}$ receives all the probability mass and the others receive zero mass?

**Summary Of The Paper:**

The paper introduces a large-scale benchmark comparison of univariate distributional time series forecasting algorithms (7 local methods; 6 global methods, and for the latter, ensembles thereof) against 44 time series datasets that have been previously introduced in the literature. It also proposes an algorithm, ParetoSelect, to choose hyperparameter defaults that lie close to the Pareto frontier in a multi-objective evaluation (e.g. model accuracy and inference latency).

**Summary Of The Review:**

The paper attempts a valuable contribution to the univariate discrete-time forecasting literature, but by attempting to cover too many things quite superficially, does not appear ready to recommend acceptance at ICLR in current form.

---

> ### Author Response · Authors · 2021-11-22
> **Response to Reviewer FZ2f [1/3]**
>
> We thank the reviewer for their comments. We address individual concerns below.
>
> **_“If viewed as a contribution to the large-scale benchmarking of time series forecasting
> algorithms, it offers a starting point but falls far short in offering substantive analysis of the
> presented results.”_**
>
> Regarding the contribution on the benchmark and its analysis, we would like to highlight the number
> of methods and datasets considered compared to previous work (see the Table in the general comment)
> and the distinctive aspect of our benchmark (accounting for multiple objectives, hyperparameters
> and seeds and ensembles). Regarding analysis of the presented results, we believe the quantitative
> statistical analysis on the amount of data required to outperform local methods is novel and will
> be valuable for the community.
>
> **_“p.5 [Comparison of classical & deep learning methods]: the paper should explain how the test
> statistic is constructed exactly. Does the test include only the raw deep learning methods, or it
> includes ensembles as well?_**
>
> We updated the manuscript to make it clearer how the test statistic is constructed (“Samples of the
> distributions are derived from the respective single-model evaluations in our benchmark”).
>
> **_“Can anything be learned from the datasets for which local methods outperform global methods
> apart from the number of examples in the dataset?”_** **_"to convey strong conclusions as to the
> suitability of particular methods for particular datasets, e.g. regarding time series properties
> such as variability, intermittency, seasonalities, etc., and possibly grouping benchmark datasets
> by broad area of study (e.g. retail, economics, energy, ...).”_**
>
> We could not identify any general patterns in the datasets where the best local method outperforms
> the best deep learning method. Based on your comments, we did, however, add a more thorough
> analysis in Appendix F and referenced it from the main text. We copy some parts of the analysis
> here:
>
> For the outlier datasets (“KDD2018”, “M3 Yearly”, “M4 Quarterly”, “Taxi”), the summary statistics
> differ wildly as can be deduced from Table 3 in the appendix. Further, the classical method that
> outperforms the best deep learning method is different for each of these datasets (“KDD2018”: NPTS,
> “M3 Yearly”: Theta, “M4 Quarterly”: ETS, “Taxi”: STL-AR). Notably, this does not even include the
> local method that performs best on average (i.e. ARIMA, see Table 1).
>
> Adding grouping information - such as the domains of time-series as suggested - as an input for a
> surrogate model predicting the performance of forecasting methods did not have much effect in
> performance. Our conclusion is that the rank of the best performing models seem to be relatively
> hard to identify only from simple dataset grouping rules. Nonetheless, we hope that by releasing
> the benchmark data, we help future research to identify dominant methods from dataset
> characteristics.
>
> **_“This approach is interesting and to the best of my knowledge is novel in a time series
> context.”_**
>
> Thank you for outlining this point. To the best of our knowledge, the method is also novel in the
> AutoML context: we are not aware of any work able to learn multi-objective defaults.
>
> **_“If viewed as contributing the ParetoSelect algorithm, then this should come much earlier in the
> paper (in its current position towards the end, it feels like an afterthought), and deeper analysis
> of the results should be presented, along with a theoretical analysis if possible.”_**
>
> We chose to introduce the ParetoSelect algorithm after the benchmark since Table 1 clearly shows
> that models have different trade-offs between accuracy and speed but it is unclear how you would
> select them which is what Pareto Select does. We added the following sentence in the introduction
> of Section 4 to better reflect this point. However, in the presence of multiple conflicting
> objectives such as accuracy and latency, it is not clear how one could choose the best models.
>
> Regarding the theoretical analysis, we added a formal guarantee (Proposition 1) that in the ideal
> case where the performance surrogate is error-free, the algorithm returns a selection with zero
> hypervolume error. We believe that investigating the case of surrogate error is an interesting
> direction of future work.

---

> ### Author Response · Authors · 2021-11-22
> **Response to Reviewer FZ2f [2/3]**
>
> **_“Section 3.4: since the forecast horizons and history length differ widely, isn't it better to
> use a relative latency measure compared to some simple benchmark (e.g. seasonal naïve)? Otherwise,
> the latency results are dominated by the datasets with the longest time series, regardless of the
> model performance per time step.”_**
>
> While we argue that the usage of the median latency alleviates the risk of the reported number
> being dominated by any one dataset, we agree that a measure of relative latency is more useful. We
> updated the manuscript accordingly and now report the relative latency compared to Seasonal Naïve,
> averaged across all datasets. Due to the suggestion of reviewer XSBL, we also include the standard
> deviation of the relative latency.
>
> **_“Section 3.5: The results reported in Table 1 include rank information across an unknown set of
> models: that's obviously more than 7+6=13 introduced in Section 3.2, and seems to include some form
> of ensembling. However, how the ensembles are constructed has not yet been introduced by that point
> in the paper.”_**
>
> This comment is certainly valid and we agree that it is not ideal to include the not-yet-introduced
> constrained ensembles here. We made the rank computation more transparent by adding a footnote that
> also includes a forward pointer to the constrained ensembles presented later.
>
> **_“p. 5: "only a few thousands observations seem sufficient to outperform the classical models
> considered": in Figure 1, there are datasets with >1e6 observations where local methods are
> indistinguishable from global methods, this is way more than "a few thousands"."_**
>
> While the formulation in the manuscript only refers to the ability of deep learning models to
> outperform local methods with few thousand observations (which is the case at ~10k observations),
> it is, in some cases, true that local methods are indistinguishable even when more than a million
> observations are available. We updated the manuscript to acknowledge this case (see second
> paragraph in the “comparison of classical and deep learning models” section). The two datasets with
> more than a million observations for which this applies are “Rideshare” and “KDD2018” (for the
> latter, the best local method also outperforms the best deep learning model). Across the whole set
> of datasets for which classical methods are indistinguishable from deep learning methods, no clear
> pattern is apparent. Again, we stress that by open-sourcing the benchmark results, more insights
> can be generated by future work.
>
> **_“p. 6 [top] it seems that $Q_deep$ and $Q_class$ represent different model results (best models
> of each class, respectively) than those given a few paragraphs above (set of all models within each
> class). It would help readability to have a more precise and coherent notation.”_**
>
> We agree with the notational conflation and updated the manuscript accordingly (see second to last
> paragraph in Section 3.5).
>
> **_“Section 4.1: it is not clear if the M different tasks represent different datasets or different
> objectives.”_**
>
> We thought the following text would clarify both variables: “an objective function maps any time
> series model $x \in X$ to $m$ objectives (such as nCRPS, latency, etc.) that ought to be minimized. In
> addition, we assume that model evaluations on $M$ different but related tasks”. Is the confusion on
> the upper/lower case use of “m”? We updated the manuscript with a different letter to avoid any
> potential confusion replacing the number of different tasks with $T$ instead of $M$.
>
> **_“And why does eq. (5) evaluates on the M related tasks only, but not on the main task? The
> notation should be tightened up.”_**
>
> Eq. (5) denotes the _offline_ evaluations used to fit the predictive model $f_\theta$, we rephrased
> to “The set of **_offline_** model evaluations is then given as”.
>
> **_“Eq (12): it's not clear how sort is defined for vectors, and does not easily relate to the
> numbering of datapoints shown in Figure 3.”_**
>
> ${\tt sort}$ is defined in Eq. 13 and the pseudo code is also given in Algorithm 1 in the appendix.
> The sort is done by a greedy algorithm (epsilon-net) that iteratively select the point that is the
> furthest way from the current selection.
>
> **_“The bibliographic entries should be checked for completeness. For instance Pfisterer et al.
> (2021) and Winkelmolen et al. (2020) are incomplete.”_**
>
> Thanks for the rigorous checking, we updated the references accordingly.

---

> ### Author Response · Authors · 2021-11-22
> **Response to Reviewer FZ2f [3/3]**
>
> **_“p. 20: "we set each quantile to the forecasted value" ==> do you mean that receives all the
> probability mass and the others receive zero mass?"_**
>
> For methods that provide point forecasts, the point forecasts can be interpreted as a probability
> distribution by casting them as a Dirac distribution centered around the forecasted value $y$. The
> Dirac distribution has infinite mass for $x = y$ and zero mass everywhere else — the CDF is a step
> function with a single step from 0 to 1 at $x = y$. Hence, the value of all quantiles is given as
> $y$. We made the explanation clearer by moving the reference to the Dirac distribution found in the
> footnote to the running text.

---

### Official Review · Reviewer_XSBL · 2021-11-03

**Correctness:** 3
**Technical Novelty And Significance:** 2
**Empirical Novelty And Significance:** 4
**Recommendation:** 6
**Confidence:** 4

**Main Review:**

**Strengths**
- This is an excellent work comparing a large and diverse set of time series forecasting approach wrt to model performance and prediction latency.
- The main contribution of this paper is empirical, presenting the latency and performance tradeoffs for various models. The study presented in this paper while not absolutely exhaustive, is still quite valuable to the community. Some ways to improve the study even further are suggested in the weaknesses section.
- The main observation made in the paper is that deep learning models are competitive or even better than classical models, and the performance gap increases only slightly with exponentially more data. This is an important observation as it breaks a common myth in the time series community. Although this statement may be a little too strong since classical models may still outperform deep models when the number of data points is less than a thousand. The datasets considered in the paper have at least a 1000 points.
- The experimental results seem convincing, and the datasets used in the paper are quite diverse and vary significantly in size, implying the robustness of the evaluation.

**Weaknesses**
- The choice of hyper-parameters (number of hidden layers, width, etc) for the various forecasting models has not been clearly mentioned. Were the models tuned in any way? or were default hyper-parameters used? This is the most significant weakness of this paper.
- Classical models may still outperform deep models when the number of data points is less than a thousand. The datasets considered in the paper have at least a 1000 points.
- It is unclear why a ranking model is used in place of a regression model. A more concrete justification should be provided for why ranking is equivalent to regression under quantile normalization.
- The argument behind using a parametric model over a non-parametric model is not completely clear. Non-parametric models such as Gaussian processes can also make use of dataset features with the right choice of a kernel. Although, GPs are not necessarily the best choice for ranking.
- The latency values in Table 1, do not include confidence intervals / stddevs (only required for the unconstrained models).


**Update after rebuttal:**
Thanks for the response. Some of my concerns have been addressed. However, I do also agree with the other reviewers that the organization of the paper and the presentation of the results can be greatly improved and hence I am keeping the same score. Details below:
- Hyper-parameters: Thanks for pointing to the appendix. Column 2 shows that different search sets for the context length have been used for different deep learning approaches. For a fair comparison, the same set of context lengths should be used since a larger context may provide an extra advantage  to the models.
- Number of data-points: This point needs to be properly addressed in the paper as well and the claims need to be re-evaluated.
- Ranking for model selection: My concerns have been fully addressed. It seems sufficiently well motivated to use parametric ranking models since the scales of the accuracy metrics can vary widely across problems. It is only feasible to rank the performance rather than predict the exact accuracy.
- Contributions in the paper: I do agree with the other reviewers about the presentation of the contributions and the organization of the paper. The fact that this paper also claims to choose default hyper-parameters for various problems, missed my attention when I was reading the paper for the first time. While the results in the paper are quite interesting, the organization can be greatly improved.

**Summary Of The Paper:**

This paper studies the tradeoff between primary model performance metrics such CRPS and CRPS rank vs other criteria such as inference latency. A surrogate model is proposed which ranks the various forecasting approaches and is used to predict the performance metrics and the prediction latency of the corresponding models. The surrogate model is used to sequentially evaluate the considered models and optimize the hyper-volume of the Pareto front. An interesting observation made in the paper is that classical models do not always beat deep methods even for relatively smaller datasets consisting of few thousand data-points. It is also shown that, as expected, there is a tradeoff between latency and performance.

**Summary Of The Review:**

Overall this a great piece of work comparing tradeoffs between multiple performance objectives for time series forecasting models, including classical and deep models. Some interesting observations have been shown in the paper which are mostly well justified. The main weakness is that the choice of hyper-parameters has not been explained, raising the concern of whether better hyper-parameter choices can improve the baseline results. I am willing to increase my score once the concerns have been addressed.

---

> ### Author Response · Authors · 2021-11-22
> **Response to Reviewer XSBL**
>
> We thank the reviewer for their comments. We address individual concerns below.
>
> **_“The choice of hyper-parameters (number of hidden layers, width, etc) for the
> various forecasting models has not been clearly mentioned. Were the models tuned in any way? or
> were default hyper-parameters used? This is the most significant weakness of this paper.”_**
>
> The hyperparameters used were not detailed in the main text due to lack of space but are detailed in
> Table 2 in the appendix. Following your comment, we added a pointer in the experiment section
> mentioning this appendix subsection.
>
> For every global model, we used the default hyperparameter setting and two other hyperparameters
> obtained by roughly doubling and halving the default model size. Additionally, we considered three
> different context lengths for all global models accounting 9 hyperparameters choice per global
> model. We also considered running more sophisticated HPO methods (e.g. random search or Bayesian
> optimization) with a larger set of choices for all models but decided against this given the
> associated price.
>
> **_“Classical models may still outperform deep models when the number of data points is less than a
> thousand. The datasets considered in the paper have at least a 1000 points.”_**
>
> We fully agree with this statement. The number of observations required for deep/global models to be outperformed
> by classical/local models is still an open question. However we believe it is highly valuable for
> practitioners to know that this value would be lower than 1000 points given that this bar is often
> achieved (and much lower than what previously thought).
>
> **_“It is unclear why a ranking model is used in place of a regression model.”_**
>
> We show in appendix D.4 that, empirically, using a list-wise ranking loss for an MLP improves the ordering of
> the model configurations significantly. We think that this can be traced back to two things: on the
> one hand, using a discounted list-wise loss function, we force the surrogate to correctly identify
> the best configurations only, paying little attention to accurately predict performances of bad
> configurations. On the other hand, we think that the ranking formulation is generally more stable
> in the presence of outliers. We rephrased this argument in the appendix to make it clearer.
>
> **_“A more concrete justification should be provided for why ranking is equivalent to regression
> under quantile normalization.”_**
>
> We regret that our formulation was not sufficiently precise, we
> did not mean to imply equivalence but to mention the fact that either ranking or regression loss
> can be used. Since we use quantile normalization, the scale of the predictive values is irrelevant.
> We updated the formulation in the paper accordingly.
>
> **_“The argument behind using a parametric model over a non-parametric model is not completely
> clear.”_**
>
> We compared with non-parametric models (kNN) in the appendix in Table 4. We used an MLP
> simply because it enables to easily incorporate a ranking objective which gives the best results.
> We did not consider GP due to the difficulty of considering ranking as you mentioned and also due
> to the potential large number of evaluations that would need to be conditioned on.
>
> **_“The latency values in Table 1, do not include confidence intervals / stddevs (only required for
> the unconstrained models).”_**
>
> In line with the suggestion by reviewer FZ2f, we replaced the
> latency column with the method’s average relative latency when compared to Seasonal Naïve.
> Following your comments, we also added the standard deviation of this relative latency, including
> for the constrained methods.

---

### Author Response · Authors · 2021-11-22
**General Response to the Reviewers**

We would like to thank all reviewers for their thorough and valuable reviews. There are two points
that we would like to highlight before answering individual comments.

First, we believe that the introduction of this benchmark is in itself a significant contribution.
We believe the amount of datasets and methods evaluated will enable to significantly improve the
reach of future empirical comparisons by allowing future researchers to benchmark for free against
a large set of methods. To illustrate this point, we listed in the following table the number of
datasets and methods used by recent forecasting papers accepted at ICLR and other top venues (where
+6 includes the 6 hyper-ensembles we benchmarked):

| Paper                                                                                                                                                                          | #Datasets | #Methods |
| ------------------------------------------------------------------------------------------------------------------------------------------------------------------------------ | -------- | -------- |
| [Multivariate Probabilistic Time Series Forecasting via Conditioned Normalizing Flows](https://openreview.net/pdf?id=WiGQBFuVRv)                                               | 6        | 7        |
| [High-Dimensional Multivariate Forecasting with Low-Rank Gaussian Copula Processes](https://proceedings.neurips.cc/paper/2019/file/0b105cf1504c4e241fcc6d519ea962fb-Paper.pdf) | 6        | 10       |
| [Deep Rao-Blackwellised Particle Filters for Time Series Forecasting](https://proceedings.neurips.cc/paper/2020/file/afb0b97df87090596ae7c503f60bb23f-Paper.pdf)               | 5        | 4        |
| [End-to-End Learning of Coherent Probabilistic Forecasts for Hierarchical Time Series](https://proceedings.mlr.press/v139/rangapuram21a.html)                                  | 5        | 6        |
| [Monash Time Series Forecasting Archive](https://arxiv.org/pdf/2105.06643.pdf)                                                                                                 | 26       | 7        |
| [Temporal Fusion Transformers for Interpretable Multi-horizon Time Series Forecasting](https://arxiv.org/pdf/1912.09363.pdf)                                                   | 4        | 9        |
| [N-BEATS: NEURAL BASIS EXPANSION ANALYSIS FOR INTERPRETABLE TIME SERIES FORECASTING](https://arxiv.org/pdf/1905.10437.pdf)                                                     | 12       | 6        |
| Ours                                                                                                                                                                           | 44       | 13+6     |

Our benchmark also has the novelty of including multiple hyperparameters, objectives and seeds and
also considering ensembles. We are not aware of any benchmark allowing to easily evaluate ensembles
for instance (even in the AutoML/NAS community). In addition to providing a large set of
evaluations that will facilitate future empirical comparisons, we believe the paper provides
important take-aways and analysis of the benchmark. For instance, the problem of the amount of data
required for deep-learning methods is vastly open (see e.g. [1]) and it gives the first formal
analysis that we are aware of.

Second, the multi-objective analysis and the contribution of ParetoSelect is more than an
after-thought in our opinion. The multi-objective aspect is a crucial, since we believe that the
practice of only reporting accuracy in time series forecasting is generally misguided. For
instance, Table 1 clearly shows that ensembling is sufficient to outperform all competitors but
that comes at a significant cost of latency. For those conflicting objectives, there is currently
no known way to learn defaults which is what our method ParetoSelect does. We would like to
highlight that it is — to the best of our knowledge — the first method able to learn defaults in a
multi-objective setting. In this sense, we believe that its simplicity is rather a strength than a
weakness especially given its ability to approximate Pareto fronts competitively in zero-shot
settings.

---

[1] Januschowski, Tim, Gasthaus, Jan, Wang, Yuyang, Salinas, David, Flunkert, Valentin,
Bohlke-Schneider, Michael, and Callot, Laurent. “Criteria for classifying forecasting methods”. In:
International Journal of Forecasting 36.1 (2020), pp. 167–177.

---

### Decision · Program_Chairs · 2022-01-20

**Decision:**

Reject

**Comment:**

The premise of this paper is that the development of time series forecasting methods has traditionally focused on accuracy rather than other criteria such as training time or latency. This work presents a new benchmark, evaluating classical and deep learning-based methods on a number of public datasets. They also propose a technique, ParetoSelect that is able to select models from the Pareto front that can efficiently select models in a multi-objective setting.

Reviewer XSBL liked the observation that classical methods do not always beat deep learning methods even for very small datasets. They thought that the empirical contribution was valuable and myth-breaking. They also commented that the evaluation was robust. Their main concerns were: inadequate description of hyperparameters, lack of evaluations on *really* small datasets, missing confidence measures for latency results. They also made some suggestions for improving clarity. The reviewers responded, pointing to a description of the hyperparameters in Table 2 of the appendix. They also responded to the reviewer’s comment about very small datasets and explained the advantage of the ranking loss. They made some small adjustments to the paper based on the clarity comments.

Reviewer PZ2f also thought that the large scale comparison was valuable for the community. Overall they thought it was well written though could be improved w.r.t. Notation and writing style. They even inspected the code. Their primary concern was that the paper lacked focus and “tries to do too much and too little”. Is this a benchmarking effort of previous methods, or is the main contribution the ParetoSelect algorithm? This reviewer thought that due to its superficial coverage of too many things, and it wasn’t ready yet for publication at ICLR. The authors provided quite a comprehensive response to reviewer PZ2f and pointed to some minor improvements in the manuscript.

Reviewer rQb3, like the others, viewed the benchmark analysis as valuable. They thought that the ParetoSelect approach was “natural” and that it was shown to be effective over baselines. Like PZ2f they had some structural criticisms and pressed for more insights.

Reviewers XSBL and rQb3 continued to engage in discussion through the AC-reviewer discussion phase. XSBL said that the authors’ response addressed some concerns yet raised others w.r.t. hyper-parameter selection. rQb3 updated their review after considering the author's response, feeling that minor concerns were addressed but the paper could still use further development. Overall, after considering the discussion I think that it’s been difficult for the authors to provide any patterns regarding which model performs best for which datasets. To me, a benchmark paper should provide some deeper insight and the paper appears to be struggling to do that. On the other hand, the study is comprehensive. The authors have argued in their response to all reviews, that their evaluation is at quite a different scale compared to other published time series model evaluations. I think that this benchmark paper can provide value to the community yet it could use further work: specifically the authors need to focus the paper and communicate clearer insights from the study.